# DELIFFAS: Deformable Light Fields for Fast Avatar Synthesis

Youngjoong Kwon[1]*, Lingjie Liu[2,3], Henry Fuchs[1],
Marc Habermann[3,4]†, Christian Theobalt[3,4]

[1]University of North Carolina at Chapel Hill. [2]University of Pennsylvania.
[3]Max Planck Institute for Informatics, Saarland Informatics Campus.
[4]Saarbrücken Research Center for Visual Computing, Interaction and AI.
{youngjoong,fuchs}@cs.unc.edu  {lliu,mhaberma,theobalt}mpi-inf.mpg.de

## Abstract

Generating controllable and photorealistic digital human avatars is a long-standing and important problem in Vision and Graphics. Recent methods have shown great progress in terms of either photorealism or inference speed while the combination of the two desired properties still remains unsolved. To this end, we propose a novel method, called DELIFFAS, which parameterizes the appearance of the human as a surface light field that is attached to a controllable and deforming human mesh model. At the core, we represent the light field around the human with a deformable two-surface parameterization, which enables fast and accurate inference of the human appearance. This allows perceptual supervision on the full image compared to previous approaches that could only supervise individual pixels or small patches due to their slow runtime. Our carefully designed human representation and supervision strategy leads to state-of-the-art synthesis results and inference time. The video results and code are available at `https://vcai.mpi-inf.mpg.de/projects/DELIFFAS`.

## 1  Introduction

Generating photorealistic renderings of humans is a long-standing and important problem in Computer Graphics and Vision with many applications in the movie industry, gaming, and AR/VR. Traditionally, creating such digital avatars from real data involves complicated hardware setups and manual intervention from experienced artists, followed by sophisticated physically-based rendering techniques to render them into an image. Thus, recent research and also this work focuses on creating a drivable and photoreal digital double of a real human learned solely from multi-view video data to circumvent the complicated and manual work.

Recent works can be categorized by their underlying scene representation. Explicit mesh-based methods represent the dynamic human by a deforming geometry and dynamic textures [15] or textured body models [45]. While these explicit methods achieve a comparably fast inference speed, the quality is still limited in terms of detail and photorealism. Hybrid approaches [14, 32] integrate the coordinate-based MLP to an explicit (potentially deforming) geometry. While their synthesis quality is drastically improved compared to explicit methods, the runtime is rather slow such that real-time is out of reach. Last, the concept of light fields [13, 38, 5] is well known for decades and recently, neural variants [47, 30, 55, 2, 52] have been proposed. Since ray-marching, *i.e.*, sampling multiple points along the ray, is not required, these methods are fast to evaluate. However, results are

---

*This work was completed during an internship at MPII.
†Corresponding author

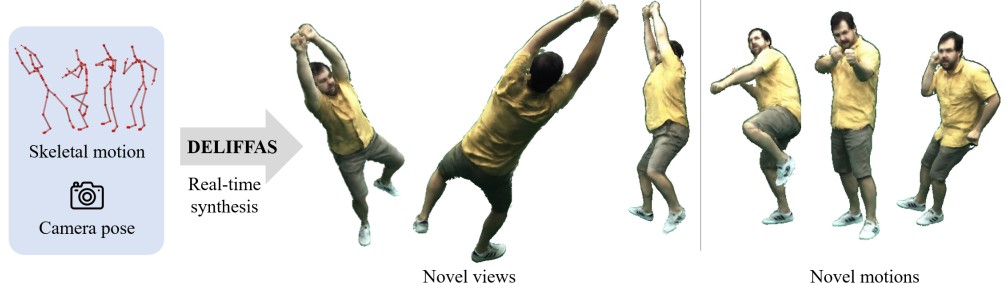

Figure 1: We present DELIFFAS, a novel method for real-time, controllable, and highly-detailed human rendering. Our method takes a skeletal motion and camera view as input and generates a corresponding photorealistic image of the actor in real-time.

only shown on static scenes. Thus, previous work either demonstrated high synthesis quality *or* fast inference speed. Our goal is to have a method, which achieves the best of the two worlds, *i.e.*, *high synthesis quality* and *fast inference speed*.

To this end, we propose *DELIFFAS*, a method for controllable, photorealistic, and fast neural human rendering. Given a multi-view video of an actor and a corresponding deformable human mesh model, which computes the deformed mesh as a function of the skeletal motion, we propose a deformable light field parameterization around the template. Typically, a light field is parameterized by a 3D position and a direction, however, this parameterization is neither robust nor efficient for deforming scenes. Therefore, we propose a deformable two-surface representation parameterizing a surface light field. In contrast to the original surface light field formulation, we deform the two surfaces according to a deformable mesh model. This allows us to efficiently query highly detailed appearance and even enables controllability since the light field can be driven by the skeletal motion. Due to the high inference speed, our method is also able to render the entire image during training, which is in stark contrast to coordinate-based approaches [32]. This allows us to employ perceptual supervision on the entire image, which is in contrast to previous works that could only supervise individual pixels or small patches due to their slow runtime. In summary, our technical contributions are:

- A novel real-time method for learning controllable and photorealistic avatars efficiently from multi-view RGB videos.
- A deformable two-surface representation parameterizing a surface light field, which allows efficient and highly accurate appearance synthesis.
- We show that our surface light field representation and our neural architecture design can be effectively parallelized and integrated into the graphics pipeline enabling real-time performance and also allowing us to employ full-image level perceptual supervision.

## 2  Related Work

**Explicit Mesh-based Methods.** Some methods use an explicit mesh template and estimate non-rigid deformations and textures on the template mesh to fit the RGB input [60, 61]. Xu et al. [62] use texture retrieval and stitching to synthesize a texture map for the novel view. Others [7, 54] propose a layered texture representation to accelerate the image synthesis. These methods present limited generalization to new poses and viewpoints. Thies et al. [53] learn deep features in the texture space. Although they can represent view-dependent effects, it cannot generalize to novel poses. To address this issue, DDC [15] leverages differentiable rendering techniques to learn non-rigid deformations and texture maps on the explicit mesh. Given novel poses and views, DDC [15] can synthesize plausible results in real time. However, our method demonstrates significantly improved visual quality while also being real-time capable. Chen et al. [9] is a single surface-based method that is tightly bound to the underlying mesh. In contrast, the proposed two-surface light field can compensate for potentially erroneous geometry and synthesizes photoreal appearance beyond the mesh bounds.

**Hybrid Methods.** The synthesis quality of explicit meshed-based methods is bounded by the limited resolution of the template mesh. On the other hand, neural implicit representations [49,

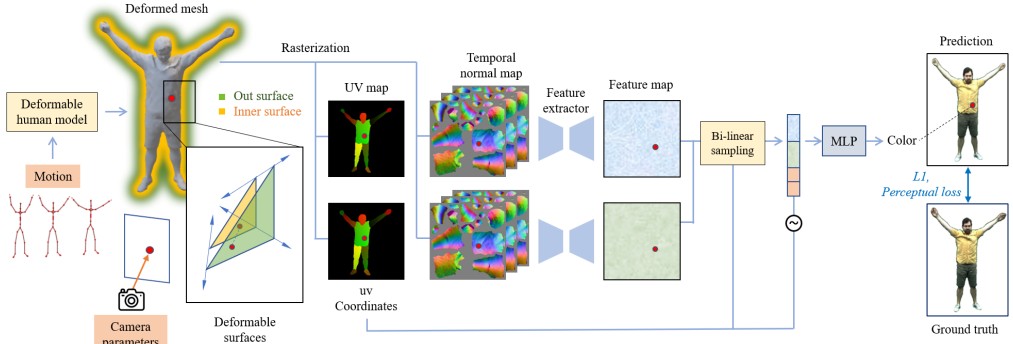

Figure 2: **Method Overview.** Given the skeletal motion and the target camera pose, we synthesize highly detailed appearance of a subject in real-time using a surface light field parameterized by our two surface representation. We first obtain the inner surface using the motion-dependent deformable human model. The outer surface is constructed by offsetting the inner surface vertice along its normal. For each camera ray, we obtain the uv coordinates of the intersecting points with the two surfaces from the image-space uv maps. Then, we bilinearly sample the features at the corresponding uv coordinates from the temporal normal feature map of each mesh. The two sampled features together with the two intersecting uv coordinates are fed into the light field MLP, which generates the color value. The rendered 2D image is supervised with $L_1$ and perceptual losses.

33, 24, 37, 31, 48] have shown their advantages over explicit representations in 3D modeling. For example, neural implicit representations are continuous, compact, and can achieve high spatial resolution. However, neural implicit representations are not animatable as deformable meshes. Therefore, recent studies propose to combine neural implicit representations with an explicit human template. Peng et al. [43] use a coarse deformable model (SMPL [34]) as a 3D proxy to optimize feature vectors, but it has limited pose generalizability. To address this issue, a large body of work [32, 41, 8, 63, 51, 39, 20, 26, 11, 66, 56, 57, 29, 50, 42, 21, 64, 27] propose neural animatable implicit representations. These works leverage the animatable 3D proxy to deform the space of different poses to a shared canonical pose space. HDHumans [14], jointly optimizes the neural implicit representation and the explicit mesh model. However, these methods suffer from slow rendering speed, taking about 5 seconds to render one frame. In contrast, our method can achieve real-time rendering ($31 fps$) and produces higher-quality synthesis.

**Light-field Methods.** The 5D plenoptic function, which represents the intensity of light observed from every position and direction in 3-dimensional space, was introduced by Bergen and Adelson [5]. Levoy et al. [28] reduce the light field to a 4D function with the two-plane parameterization, *i.e.*, light slab. Wood et al. [58] propose a surface light field, which maps a point on the base mesh and viewing direction to the radiance. While they can parameterize a full 360-degree ray space, they cannot go beyond the underlying mesh and can only represent a static scene. Recently, learning-based light field approaches have been proposed. Convolutional neural network-based methods [23, 4, 36] learn to interpolate and extrapolate the light field from a sparse set of views. Some works [30, 55, 2] achieve high quality synthesis with the light field represented as an MLP. However, they are limited to the fronto-parallel scene synthesis. To represent the full 360-degree ray space, some works [47, 52] utilize Plücker and two-sphere parameterization [18, 6], respectively. However, none of the aforementioned works can represent the full $360°$ light space of a dynamic scene. Ouyang et al. [40] leverages multi-plane image and perceptual supervision to render a photorealistic avatar in real-time. However, they only allow minimal pose changes. In our work, we introduce a deformable two-surface representation parameterizing the surface light field, which enables us to represent and control the dynamic scene, *i.e.*, the human, and to handle arbitrary poses during inference.

## 3 Method

Our goal is to learn an animatable 3D human character from multi-view videos, which can be rendered photorealistically from novel views and in novel motions in real time. In this endeavor, we propose a novel method, called DELIFFAS, which at inference takes a skeletal motion sequence and a target

camera pose as input and renders a high-quality image of the subject in the specified pose under the virtual camera view at *real-time* framerates. An overview of our approach is shown in Fig. 2. At the core, our method represents the digital human as a surface light field, which is parameterized by two deforming surfaces that are directly controlled through the skeletal motion. Next, we provide some background concerning our data assumptions, deformable surface representations, and light fields (Sec. 3.1). Then, we introduce our deformable surface light field parameterized by a motion-dependent deformable two-surface representation and how it can be efficiently rendered (Sec. 3.2). Lastly, we introduce our perception-based supervision strategy (Sec. 3.3).

## 3.1 Background

**Data Assumptions.** We assume a segmented multi-view video of the actor, who is performing a diverse set of motions, is given using $C$ calibrated and synchronized cameras. Moreover, we also assume the skeletal motion $\boldsymbol{\theta}_f \in \mathbb{R}^D$ for a frame $f$ of the video can be obtained, *e.g.*, by using markerless motion capture. Here, $D$ denotes the number of degrees of freedom of the skeleton.

**Skeletal Motion-dependent Surface.** In this work, we mainly focus on the question of how to efficiently render a person at a photorealistic quality and less on surface reconstruction or modeling. Thus, we assume a skeletal motion-dependent deformable mesh surface $s(\boldsymbol{\theta}_{f,f-T}) = \mathbf{V}_{\boldsymbol{\theta}_f} : \mathbb{R}^{T \times D} \to \mathbb{R}^{N \times 3}$ of the human is given. Here, $\boldsymbol{\theta}_{f,f-T}$ denotes the motion window from $f-T$ to the current pose $f$ and $\mathbf{V}_{\boldsymbol{\theta}_f}$ denotes the deformed and posed mesh vertex positions at frame $f$ with pose $\boldsymbol{\theta}_f$. $N$ is the number of vertices. In practice, we leverage the deformable human model of Habermann et al. [16]. However, any other deformable surface representation such as SMPL [34] could be used as well (see Sec. D for more discussions).

**Light Field.** The light field [13, 38, 5] is a function $l(\mathbf{x}, \mathbf{d}) = \mathbf{c}$ that maps an oriented camera ray, here parameterized by its origin $\mathbf{x} \in \mathbb{R}^3$ and the direction $\mathbf{d} \in \mathbb{R}^3$, to the transmitted outgoing radiance along the camera ray. Note that only a single evaluation per pixel is required when rendering the light field into a discrete image, which is in stark contrast to the recently proposed Neural Radiance Fields [37] that require hundreds of computationally expensive network evaluations along the ray. Thus, light fields offer a computationally efficient alternative. Among the many possible ways of parameterizing a ray, the two-plane parameterization [28], *i.e.*, light slab, is the most commonly used method. This two-plane method parameterizes a ray by its intersections with two planes $\mathcal{U}_1$ and $\mathcal{U}_2 \in [0, 1]^2$. Without loss of generality, we here define the planes being contained in a range from zero to one and assume the field is always first intersecting $\mathcal{U}_1$ and then $\mathcal{U}_2$. Thus, the light field formulation can be adapted to

$$l(\mathbf{u}_1, \mathbf{u}_2) = \mathbf{c} \tag{1}$$

where $\mathbf{u}_1 \in \mathcal{U}_1$ and $\mathbf{u}_2 \in \mathcal{U}_2$ are the intersection points of a ray with the planes. However, it is hard to model full $360°$ view changes with this light slab representation, and it originally was not designed for controlling the scene, in our case the human. Thus, in the remainder of this section we demonstrate how this classical concept can be extended to our setting and how our formulation enables fast *and* photoreal renderings of humans.

## 3.2 Deformable Light Field Parameterization

As discussed above, one major challenge for the traditional two-plane representation is that it cannot cover the full $360°$ in terms of viewing directions. Thus, we first explain how we adopt this representation using the deformed character mesh.

**$360°$ Viewpoints.** Since we have the deformed mesh $\mathbf{V}_{\boldsymbol{\theta}_f}$, our idea is to adapt the two-plane parameterization to a two-(non-planar)-surface parameterization using the deformed mesh. While the inner surface (equivalent to $\mathcal{U}_2$) can be represented by the original mesh $\mathbf{V}_{\boldsymbol{\theta}_f}$ (denoted as $\mathbf{V}_{\boldsymbol{\theta}_f}^{\text{inner}}$ in the remainder), we have to construct an outer surface $\mathcal{S}_1$ for which we can guarantee that every ray first intersects $\mathcal{S}_1$ before intersecting $\mathcal{S}_2$. This can be achieved by constructing an offset surface as

$$\mathbf{V}_{\boldsymbol{\theta}_f, i}^{\text{out}} = \mathbf{V}_{\boldsymbol{\theta}_f, i}^{\text{inner}} + d \cdot \mathbf{n}_i^{\text{inner}} \tag{2}$$

where $i$ denotes the $i$-th vertex, $\mathbf{n}_i^{\text{inner}}$ is the respective normal on the inner surface, and $d$ defines the length of the offset. We use d=3cm in the experiments. By offsetting all vertices, we obtain the outer surface $\mathbf{V}_{\boldsymbol{\theta}_f}^{\text{out}}$. By assuming the mesh is watertight and that the origin $\mathbf{o}$ of a ray with direction $\mathbf{d}$ lies

outside of $\mathbf{V}_{\boldsymbol{\theta}_f}^{\text{out}}$, we can guarantee that this ray will always intersect $\mathbf{V}_{\boldsymbol{\theta}_f}^{\text{out}}$ before $\mathbf{V}_{\boldsymbol{\theta}_f}^{\text{inner}}$. Note that we can now have plausible intersections with the two surfaces from (almost) any point in 3D space enabling 360° viewpoints, which are typically difficult for the classical two-plane parameterization.

**2D Surface Parameterization.** Now if the ray intersects the outer and inner surface at $\mathbf{p}^{\text{out}}$ and $\mathbf{p}^{\text{inner}}$, respectively, we are interested in converting it into the original light slab parameterization (Eq. 1) that solely consists of two 2D coordinates. We achieve this by performing UV mapping, *i.e.*, we construct a texture atlas $m(\mathbf{p}) = \mathbf{w}$ for the original mesh $\mathbf{V}_{\boldsymbol{\theta}_f}^{\text{inner}}$. $m$ maps a point $\mathbf{p}$ on the 3D surface to a 2D location $\mathbf{w} \in [0,1]^2$ on a plane. Note that the atlas remains the same across poses since the connectivity of the mesh is fixed. Further, our offset surface construction also preserves the texture atlas and, thus, the atlas for the outer surface is equivalent to the inner one. Now, our 3D intersection points $\mathbf{p}^{\text{out}}$ and $\mathbf{p}^{\text{inner}}$ can be mapped to 2D as $m(\mathbf{p}^{\text{out}}) = \mathbf{w}^{\text{out}}, m(\mathbf{p}^{\text{inner}}) = \mathbf{w}^{\text{inner}}$ and the light slab formulation can be adapted as

$$l(\mathbf{w}^{\text{out}}, \mathbf{w}^{\text{inner}}) = \mathbf{c}. \tag{3}$$

Note that both 2D coordinates are not dependent on the motion of the character and, therefore, the light field $l$ cannot model motion-dependent appearance. Next, we further refine our formulation such that $l$ can potentially model this.

**Skeletal Motion-dependent Conditioning.** We note that the human mesh $\mathbf{V}_{\boldsymbol{\theta}_f}^{\text{inner}}$ is already a function of skeletal motion, though it is not in a format for efficient encoding considering the light slab formulation. We convert the inner and outer surface into temporal normal maps [15] $\mathcal{T}_{\boldsymbol{\theta}, f:f-T}^{\text{inner}}$ and $\mathcal{T}_{\boldsymbol{\theta}, f:f-T}^{\text{out}}$, which are the concatenation of the posed normal maps of the motion window $[f, f-T]$. Note that both maps encode information about the skeletal motion and can be directly generated by the motion-dependent mesh surface $s(\cdot)$.

Now, we deeply encode them into feature maps $f_\Omega(\mathcal{T}_{\boldsymbol{\theta}, f:f-T}^{\text{inner}})$ and $f_\Psi(\mathcal{T}_{\boldsymbol{\theta}, f:f-T}^{\text{out}})$ with a channel size of 32 respectively using deep convolutional networks $f$ with weights $\Omega$ and $\Psi$. Last, our light slab formulation (Eq. 3) can be again refined as

$$l(\gamma(\mathbf{w}^{\text{out}}), \gamma(\mathbf{w}^{\text{inner}}), \Pi(f_\Psi(\mathcal{T}_{\boldsymbol{\theta}, f:f-T}^{\text{out}}), \mathbf{w}^{\text{out}}), \Pi(f_\Omega(\mathcal{T}_{\boldsymbol{\theta}, f:f-T}^{\text{inner}}), \mathbf{w}^{\text{inner}})) = \mathbf{c}. \tag{4}$$

Here, $\Pi$ denotes the bilinear sampling operator. $\gamma$ is the positional encoding [37]. Intuitively, the light slab $l$ takes the bilinearly sampled features on the UV plane at the uv coordinates of the intersections points of the ray with the respective plane. Now, our light slab formulation is not only view-dependent but is also motion-dependent since the features are encoding the skeletal motion.

In case the ray is not intersecting the inner surface, inspired by depth peeling, uv coordinates and features are sampled at the second intersection with the outer surface. Here, the watertightness of the model guarantees that such a second intersection point exists. We note that our light slab $l$ is represented as an 8-layer MLP with 256 channel-size.

**Efficient Rendering.** Next, we explain how the above formulation can be efficiently computed using the standard graphics pipeline and deep learning tools. First, the two convolutional networks $f_\Psi$ and $f_\Omega$ have to encode the respective normal maps. The computational effort is independent of the number of foreground pixels, *i.e.*, pixels which are covered by the subject. Next, for each pixel we have to obtain the uv coordinates of the intersection points with the inner and outer surface. Here, standard GPU-parallelized rasterization can be leveraged by rendering the uv coordinates into screen space. The only computation that linearly scales with the number of foreground pixels is the evaluation of the light slab $l$, but since there is only one evaluation per ray, also this step is computationally efficient leading to an inference time of $31 fps$.

### 3.3 Supervision and Training Procedure

We supervise our approach by minimizing the following loss $\mathcal{L} = \lambda_1 \cdot \mathcal{L}_1 + \lambda_{\text{perc}} \cdot \mathcal{L}_{\text{perc}}$, where $\mathcal{L}_1$ and $\mathcal{L}_{\text{perc}}$ are the $L_1$ loss and perceptual [22] loss, respectively. $\lambda_1$ and $\lambda_{\text{perc}}$ are their weights. We use the VGG-16 network [46] pre-trained on ImageNet [10] to compute the perceptual similarity loss. Although our method solely supervised with the $L_1$ loss already outperforms other real-time methods and most of the non-real time methods on the novel view synthesis task (see Tab. 1a and Tab. 2-c), it is still blurry. This is due to the one-to-many mapping problem [3, 32] where the single skeletal pose can still induce various different appearances. Thus, the perceptual loss [22] is additionally employed

| Method | PSNR↑ | LPIPS↓ | FID↓ | Real-time |  | Method | PSNR↑ | LPIPS↓ | FID↓ | Real-time |
|---|---|---|---|---|---|---|---|---|---|---|
| NB | 29.94 | 42.15 | 109.98 | ✗ |  | NB | **29.37** | 43.99 | 115.70 | ✗ |
| A-NeRF | 29.54 | 35.27 | 86.90 | ✗ |  | A-NeRF | 28.42 | 38.74 | 95.56 | ✗ |
| NA | 30.21 | 23.60 | 18.56 | ✗ |  | NA | 28.43 | 28.33 | 24.50 | ✗ |
| HDHumans | **31.00** | **14.61** | **4.93** | ✗ |  | HDHumans | 27.69 | **24.00** | **9.25** | ✗ |
| NHR | 28.39 | 46.07 | 116.59 | ✓ |  | NHR | **28.08** | 47.65 | 122.60 | ✓ |
| NV | 25.49 | 85.69 | 123.19 | ✓ |  | NV | 23.32 | 98.35 | 139.82 | ✓ |
| DDC | 32.96 | 20.07 | 27.73 | ✓ |  | DDC | 28.05 | 30.43 | 38.37 | ✓ |
| **Ours** | **33.30** | **12.85** | **8.69** | ✓ |  | **Ours** | 27.95 | **26.59** | **26.16** | ✓ |

(a) *View synthesis error* on *D2*          (b) *Motion synthesis error* on *D2*

Table 1: **Quantitative results** on novel view and pose synthesis. (a) Our method achieves the best performance among all the real-time and non-real time methods in terms of PSNR and LPIPS, and the second best performance on FID. (b) Our method again achieves high performance on the perceptual metrics while running in real-time. While HDHumans has slightly better results in terms LPIPS and FID compared to our work, their method is $\times 100$ slower than our method.

and this improves the fine-grained details as can be seen in the ablation study (see Fig. 5-c,d). In contrast to patch-based perceptual supervision methods [44, 35, 57, 12], our fast rendering speed enables employing the perceptual supervision on *the entire image*.

## 4 Results

DELIFFAS runs at $31 fps$ when rendering a $1K$ ($940 \times 1285$) video using a single A-100 GPU with an Intel Xeon CPU. The implementation details and additional results are provided in the appendix.

**Dataset.** We evaluate our method on the DynaCap dataset [16], which is publicly available. DynaCap provides performance videos of 5 different subjects captured from 50 to 101 cameras, foreground mask, and skeletal pose corresponding to each frame, and the template mesh for each subject. The training and testing videos consist of around $20k$ and $5k$ frames at the resolution of $1K$ ($1285 \times 940$), respectively. Four cameras are held out for the testing and the remaining ones are used for the training as proposed by the original dataset. Among the available subjects, we choose $D_1$, $D_2$, and $D_5$ subjects for our experiments, which vary in terms of clothing style, *i.e.*, loose skirts and trousers.

**Metrics.** We evaluate our performance using the peak signal-to-noise ratio (PSNR). However, human perception cannot be fully reflected with this metric since a very blurry and unrealistic result can still lead to a low error [65]. Therefore, we additionally employ the learned perceptual image patch similarity (LPIPS) [65], and the Fréchet inception distance (FID) [17], which are similar to the human perception. We generate and evaluate results at $1K$ resolution ($1285 \times 940$). Every metric is computed by averaging across the entire sequence using every $10th$ frame. Four held-out cameras (7, 18, 27, 40), which are uniformly sampled in the space, are used for the testing.

### 4.1 Qualitative Results

We first present the qualitative results on the novel view synthesis task in Fig. 3-(A) and on the novel pose synthesis task in Fig. 3-(B). For both tasks, we show the synthesis result of the subject in two different poses viewed from two different viewpoints. Our method synthesizes images with fine-scale details including the facial features and wrinkles of the clothing for both tasks. Note that our method can even generate plausible results of the subject in a challenging garment type, *i.e.*, loose skirt. Furthermore, we can synthesize high-quality renderings of the subject under difficult testing poses. These results confirm the versatility as well as the generalization ability of our method.

### 4.2 Comparison

We compare our method with the state-of-the-art methods concerning novel view and pose synthesis. We choose baselines from four different classes, which are volume rendering, explicit mesh, hybrid, and point cloud-based methods. For the volume rendering-based method, we compare with Neural Volumes (NV) [33], which learns a volumetric representation. Deep Dynamic Character (DDC) [16] is an explicit mesh-based method, where the network learns to render the neural texture obtained by rasterizing the explicit mesh. Neural Actor (NA) [32], Neural Body (NB) [43], and A-NeRF [51] are hybrid methods that combine the human prior, *i.e.*, naked human body model [34], and skeleton,

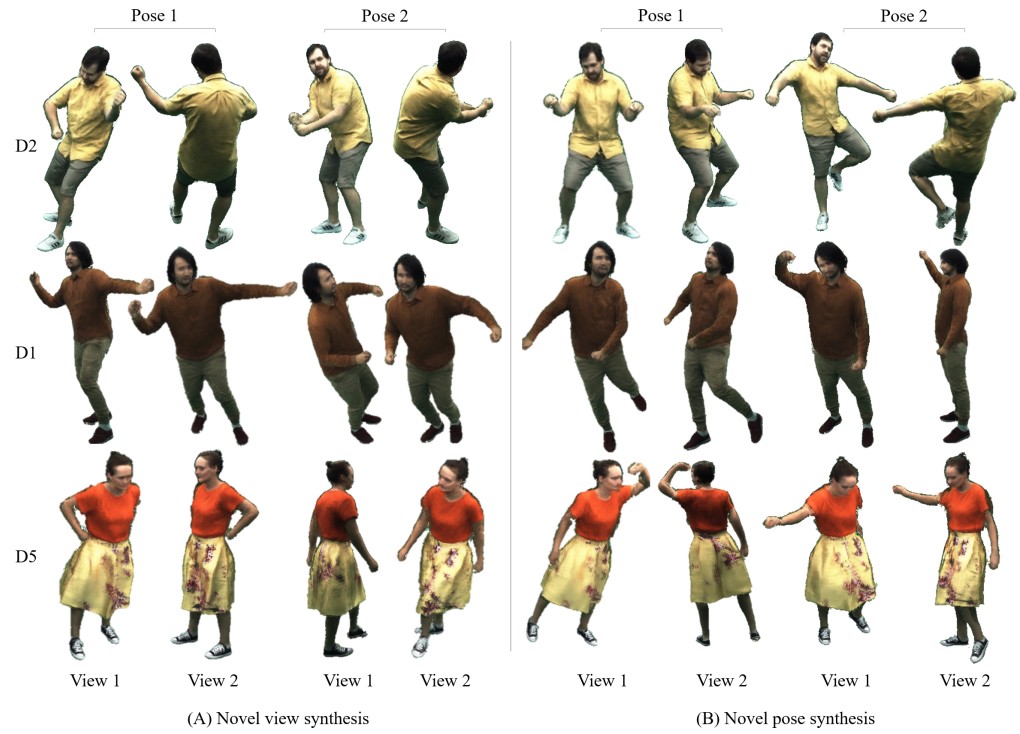

(A) Novel view synthesis                    (B) Novel pose synthesis

Figure 3: **Qualitative results** for novel view and poses. Our method achieves photorealistic rendering quality while running in real time. We are able to recover the fine-grained details including the facial features and clothing wrinkles. The high-quality synthesis results on the very challenging loose garment, *i.e.*, skirt, and the complicated testing poses show the versatility of our method.

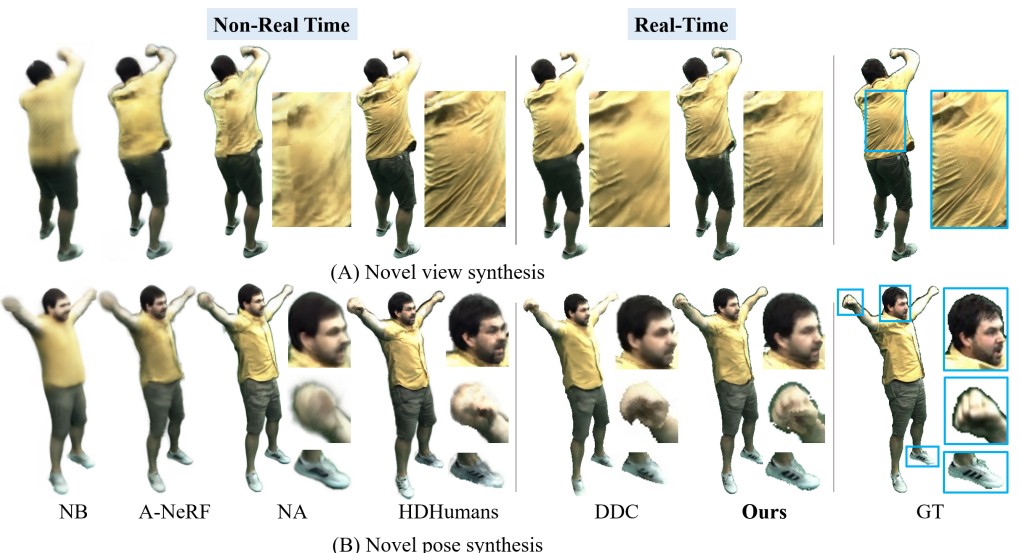

Figure 4: **Comparisons** on the novel view and pose synthesis task on the $D_2$ subject. Our method synthesizes realistic and delicate details in real time ($31 fps$). Note that we achieve similar or better quality as offline approaches and show drastically improved quality compared to real-time methods.

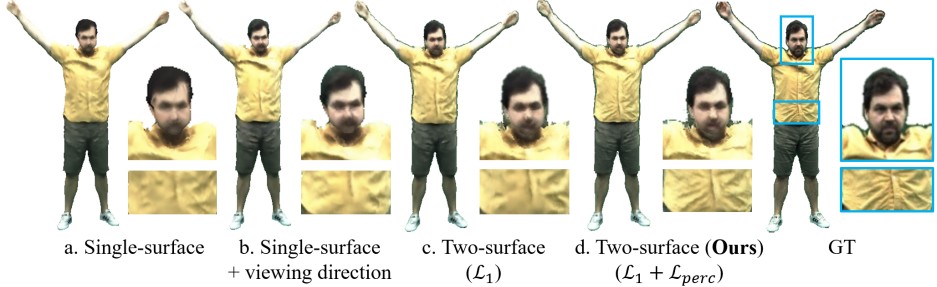

| a. Single-surface | b. Single-surface
+ viewing direction | c. Two-surface
$(\mathcal{L}_1)$ | d. Two-surface (**Ours**)
$(\mathcal{L}_1 + \mathcal{L}_{perc})$ | GT |

Figure 5: **Ablation** of each design choice on the novel view synthesis task. All the variants except (d) are trained only with $L_1$ loss. Although conditioning on the viewing direction (b) improves the quality, single-surface models (a,b) are tightly bounded to the underlying mesh. On the other hand, our proposed method (c,d) can go beyond the mesh and recover the real geometry. Additional utilization of the perceptual loss (d) improves the level of detail.

with a NeRF. Neural Actor (NA) achieves high-quality results by utilizing the adversarial supervision in texture space. Neural Human Renderer (NHR) [59] optimizes features that are anchored to point clouds. We additionally compare to the concurrent work HDHumans [14], which is a hybrid method that combines the deformable human model [16] with a NeRF. Similar to NA, HDHumans also employs adversarial supervision in texture space. We show our comparison results on the $D_2$ subject with tight clothing, as other baselines except DDC and HDHumans cannot handle the loose garment types. Note that we present both the qualitative and quantitative results in the order of *non-real time*, *i.e.*, NB, A-NeRF, NA, HDHumans, and *real-time* methods, *i.e.*, NHR, NV, DDC, and ours.

Fig. 4-(A) shows the comparison on the novel view synthesis task. Different from their original results [43], NB produces blurry results when trained on longer and more challenging training sequences, *e.g.*, around 20k frames, as they fail to optimize the appearance code for every frame. A-NeRF exhibits the loss of details under highly articulated poses. While DDC and NA can generate coarse details, they still lack high-frequency details. On the other hand, our method can recover, both, the coarse and fine details and only achieves slightly worse synthesis quality than HDHumans while being significantly faster ($\times100$). In Tab. 1a, our method outperforms all the non-real time and real-time methods except HDHumans. However, we would like to highlight again that HDHumans has a runtime in the order of seconds per frame, which prevents it from being used in any real-time application. It is worth noting that ours with $L_1$-only supervision (see Tab. 2-c) already outperforms other real-time methods and even the non-real time method NA.

Fig. 4-(B) shows the comparisons on the novel pose synthesis task. Again, our method can recover the sharp and high-frequency details that are close to the ones of HDHumans, while most of the other approaches struggle with the given challenging pose. This is consistent with the quantitative results in Tab. 1b, where we achieve the second best performance among all the methods and the best performance among the real-time methods on the perception-based metrics LPIPS and FID. Due to the innate nature of pose to appearance task, *i.e.*, the one-to-many mapping [3, 32], our method can generate realistic looking details that can deviate from the ground truth, which explains the lower performance in terms of PSNR in Tab. 1b.

### 4.3 Ablation

We perform ablation studies on the novel view synthesis task on the $D_2$ subject. To accurately access the contribution of each component, we train all the variants only with $L_1$ loss if not stated otherwise.

**Impact of Two-surface Parameterization.**  To study the efficacy of the two-surface parameterization, we train the single-surface model ('single-surface') where the MLP generates color conditioned only on a single inner surface intersection coordinates and feature. We also train a single-surface variant that is additionally conditioned on the viewing direction ('single-surface + viewing direction'). In Tab. 2, our two-surface model (c) outperforms the single-surface variants (a,b). Also, we would like to again stress that our $L_1$-only supervised two-surface model (c) already surpasses all of the real-time methods including DDC, and also outperforms the non-real time method NA, which is trained with ground truth texture-map supervision and adversarial supervision. This confirms that the

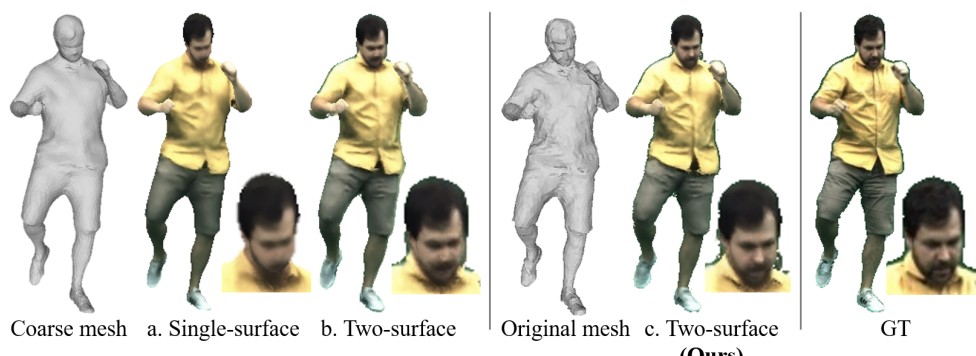

Coarse mesh  a. Single-surface  b. Two-surface  |  Original mesh  c. Two-surface  GT
**(Ours)**

Figure 6: **Ablation on the deformable template quality.** All baselines trained only with $L_1$ loss. When using a coarse mesh model the single-surface baseline (a) suffers from the geometric error of the model. In contrast, our two-surface representation (b) has the capability of fixing geometric errors and shows similar quality compared to the result when using the original mesh model (c).

| Method | | PSNR↑ | LPIPS↓ | FID↓ |
|---|---|---|---|---|
| a. | Single-surface | 31.34 | 22.74 | 28.96 |
| b. | Single-surface + viewing direction | 32.19 | 20.38 | 23.88 |
| c. | Two-surface with $L_1$ | 33.27 | 18.51 | 23.58 |
| d. | Two-surface with $L_1 + \mathcal{L}_{\text{perc}}$ (**Ours**) | **33.30** | **12.85** | **8.69** |
| e. | Single-surface with coarse mesh | 29.63 | 27.75 | 42.84 |
| f. | Two-surface with coarse mesh | 32.03 | 22.00 | 31.43 |

Table 2: **Ablations** on novel view synthesis for the $D_2$ subject. All baselines except (d) are trained with $L_1$-only supervision. Our two-surface parameterization outperforms single-surface variants and the perceptual supervision further improves visual quality. Further, our results when using a coarse mesh model are still plausible demonstrating the robustness of our method.

improvement originates from the carefully-designed two-surface model. In Fig. 5, We visually ablate the impact of each design choice. Although conditioning on the viewing direction (b) improves the quality, the single-surface models (a,b) are tightly bound to the underlying mesh. On the other hand, our two-surface models (c,d) go beyond the mesh and recover the real geometry.

**Impact of Perceptual Supervision.** Perceptual supervision improves the details (*e.g.*, wrinkles) in Fig. 5-d, which is also confirmed by the better performance in the perceptual metrics in Tab. 2-d.

**Impact of Deformable Template Quality.** To demonstrate the robustness of our method, we train and evaluate our method with a smoothed version of the original mesh model in Fig. 6 and Tab. 2-e,f. The quantitative result outperforms real-time methods including DDC in terms of PSNR. This shows that even coarser geometry is sufficient for our representation. Moreover, the single-surface model (Fig. 6-a) reveals and suffers from geometric error when the coarse template is used. On the other hand, our two-surface representation (Fig. 6-b) has the capability of fixing geometric errors and shows similar visual quality to the result when the original mesh model is used (Fig. 6-c).

## 5 Conclusion

We presented DELIFFAS, an animatable representation that can synthesize high-quality images of the avatar under a user-controlled skeletal motion and viewpoint in real-time. At the technical core, our method utilizes a surface light field parameterized with two deformable surfaces computed from a deformable human model. Then, our light slab queries pose-dependent temporal normal map features together with the intersecting uv coordinates in order to render the pixel color with just a single MLP evaluation per pixel. This efficient rendering allows perceptual supervision on the entire image. Our experiments show that the proposed representation can synthesize high-quality renderings even under challenging poses, and outperforms the state-of-the-art real-time methods.

**Acknowledgments.** Christian Theobalt and Marc Habermann were supported by ERC Consolidator Grant 4DReply (770784). Lingjie Liu was supported by Lise Meitner Postdoctoral Fellowship. This work was partially supported by National Science Foundation Award 2107454.

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

## A  Appendix - Overview

This appendix is organized as follows: Sec. B shows additional results including video results, ablations on image-level and patch-level perceptual supervision, ablation study with and without perceptual supervision, ablations on the single surface with the same network capacity, results when using SMPL, comparison with traditional multi-view stereo method; Sec. C provides information regarding the reproducibility, which includes implementation details, training details, and runtime at inference; Sec. D discusses the performance changes according to the template mesh quality, and possible alternatives for the template mesh; Sec. E presents the societal impacts our work can have; Sec. F discusses the limitations of this work.

## B  Additional Results

### B.1  Video Results

Video results of free-viewpoint renderings, novel view and pose synthesis, and comparison with the state-of-the-art baselines on the DynaCap dataset [16] can be found at `https://vcai.mpi-inf.mpg.de/projects/DELIFFAS`. We compare our DELIFFAS with the non-real-time hybrid method Neural Actor [32], and real-time explicit mesh-based method Deep Dynamic Characters [16].

### B.2  Ablation on Image-level and Patch-level Perceptual Supervision

To study the impact of our full image-level supervision, we train a variant with patch-level perceptual supervision. Here, we use a $32 \times 32$ patch. In Tab. 3, our approach trained with full-image-level perceptual supervision (Tab. 3-c) outperforms the patch-level variant (Tab. 3-b) in terms of the perceptual metrics. In Fig. 7, we ablate the visual impact. Employing full-image-level supervision leads to better reconstruction of details (*e.g.*, wrinkles). We would like to again highlight that image-level supervision with perceptual loss is possible thanks to the fast rendering speed of our method.

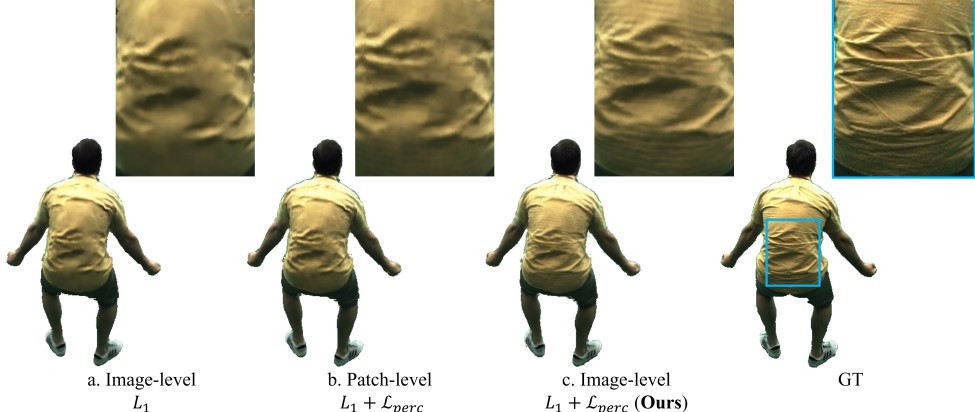

|  a. Image-level  |  b. Patch-level  |  c. Image-level  |  GT  |
|  $L_1$  |  $L_1 + \mathcal{L}_{perc}$  |  $L_1 + \mathcal{L}_{perc}$ (**Ours**)  |  |

Figure 7: **Ablation** of patch-based and image-level perceptual supervision. Variants with the perceptual supervision (b,c) shows better details than the result with $L_1$-only supervision (a). Our final model which employs the full image-level perceptual supervision (c) outperforms the variant with patch-level supervision (b).

### B.3  Ablation Study with and without Perceptual Supervision

We excluded the perceptual supervision from the ablation study in the main text (see Tab. 2, Fig. 5, and Fig. 6) to verify the effectiveness of our two-surface design. In Tab. 4 and Fig. 8, we show the ablation results with perceptual supervision. Similar to the results without perceptual supervision, our full model outperforms other variants.

| Method | PSNR↑ | LPIPS↓ | FID↓ |
|---|---|---|---|
| a.  Image-level with $L_1$ | 33.27 | 18.51 | 23.58 |
| b.  Patch-level with $L_1 + \mathcal{L}_{\text{perc}}$ | 33.24 | 16.54 | 18.96 |
| c.  Image-level with $L_1 + \mathcal{L}_{\text{perc}}$ (**Ours**) | **33.30** | **12.85** | **8.69** |

Table 3: **Ablation** on image-level and patch-level supervision. Applying the perceptual supervision on the entire image (c) leads to better results than the patch-level perceptual supervision (b) or $L_1$-only supervision.

| Method | PSNR↑ | LPIPS↓ | FID↓ |
|---|---|---|---|
| $L_1$ | | | |
| a.  Single-surface | 31.34 | 22.74 | 28.96 |
| b.  Single-surface + viewing direction | 32.19 | 20.38 | 23.88 |
| c.  Single-surface + two U-Net | 31.47 | 15.43 | **19.99** |
| d.  Two-surface (**Ours**) | **33.27** | **18.51** | 23.58 |
| e.  Single-surface with coarse mesh | 29.63 | 27.75 | 42.84 |
| f.  Two-surface with coarse mesh | 32.03 | 22.00 | 31.43 |
| $L_1 + L_{perc}$ | | | |
| g.  Single-surface | 31.36 | 16.53 | 10.90 |
| h.  Single-surface + viewing direction | 31.95 | 15.96 | 9.01 |
| i.  Single-surface + two U-Net | 31.26 | 16.21 | 9.61 |
| j.  Two-surface (**Ours**) | **33.30** | **12.85** | **8.69** |
| k.  Single-surface with coarse mesh | 29.55 | 20.05 | 15.22 |
| l.  Two-surface with coarse mesh | 32.03 | 16.13 | 12.08 |

Table 4: **Ablations** on novel view synthesis for the $D_2$ subject **with and without perceptual supervision**. Our two-surface parameterization performs better than the single-surface variants and the perceptual supervision further boosts the visual quality. In addition, our results with coarse mesh model shows reasonable performance, which confirms the robustness of our method.

## B.4   Ablations on the Single Surface with the Same Network Capacity

The number of total parameters is larger in the two-surface model than in the single-surface model as it includes two separate U-Nets for extracting feature map from each surface while the single-surface model only has a single U-Net. The U-Net and MLP architecture used in both single and two-surface models are the same. To verify that the performance boost originates from the two-surface design rather than the increase in network capacity, we additionally trained a "single-surface with two concatenated U-Net" variant (see Fig. 9, Tab. 4-c,i). Even though now the network size is the same, our two-surface representation still outperforms the single-surface baseline.

## B.5   Using SMPL as the Deformable Mesh Surface

We conduct an experiment where we use the SMPL model as the deformable mesh model (see Fig. 10). One can see that our approach even for this very coarse model still outperforms the baselines proving the flexibility of our representation while a better template mesh further improves the results.

## B.6   Comparison with Traditional Multi-view Stereo Method

We compare with the commercial photogrammetry software, Metashape [1] in Fig. 11. The reconstructed surface is noisy and the texture suffers from ghosting artifacts due to the inaccurate geometry. Also, Metashape takes 6 minutes to reconstruct while ours works in real-time ($31 fps$). Lastly, our method is animatable, while multi-view stereo methods are not.

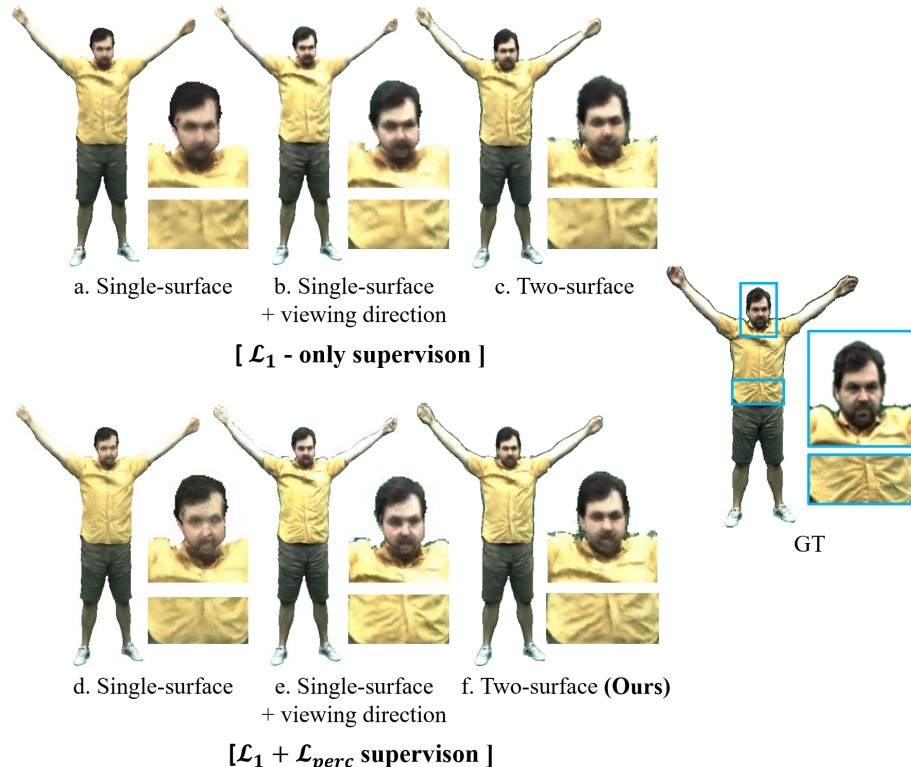

a. Single-surface     b. Single-surface     c. Two-surface
                + viewing direction

**[ $\mathcal{L}_1$ - only supervison ]**

d. Single-surface     e. Single-surface     f. Two-surface **(Ours)**
                + viewing direction

**[$\mathcal{L}_1 + \mathcal{L}_{perc}$ supervison ]**

GT

Figure 8: **Ablations** on novel view synthesis for the $D_2$ subject **with and without perceptual supervision**. The proposed two-surface representation helps to recover the real geometry while the single-surface design is limited to the underlying mesh. Perceptual supervision further enhances the quality.

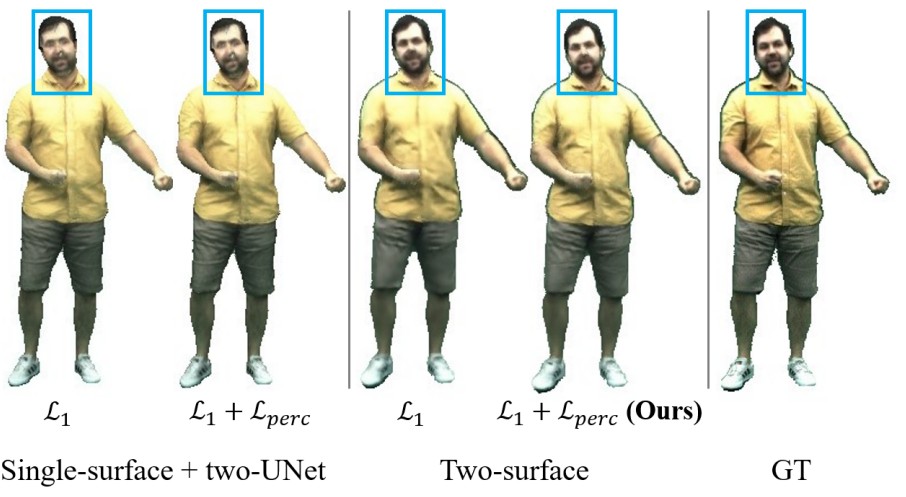

$\mathcal{L}_1$        $\mathcal{L}_1 + \mathcal{L}_{perc}$        $\mathcal{L}_1$        $\mathcal{L}_1 + \mathcal{L}_{perc}$ **(Ours)**

Single-surface + two-UNet        Two-surface        GT

Figure 9: **Ablations** on the single surface with **the same network capacity**. Evaluated on the novel view synthesis task for the $D_2$ subject. Even though the network size of the "single-surface with two U-Net" variant and ours are the same, our two-surface representation still outperforms the single-surface baseline. This again verifies that the performance boost is coming from the two-surface design, and not from using more network.

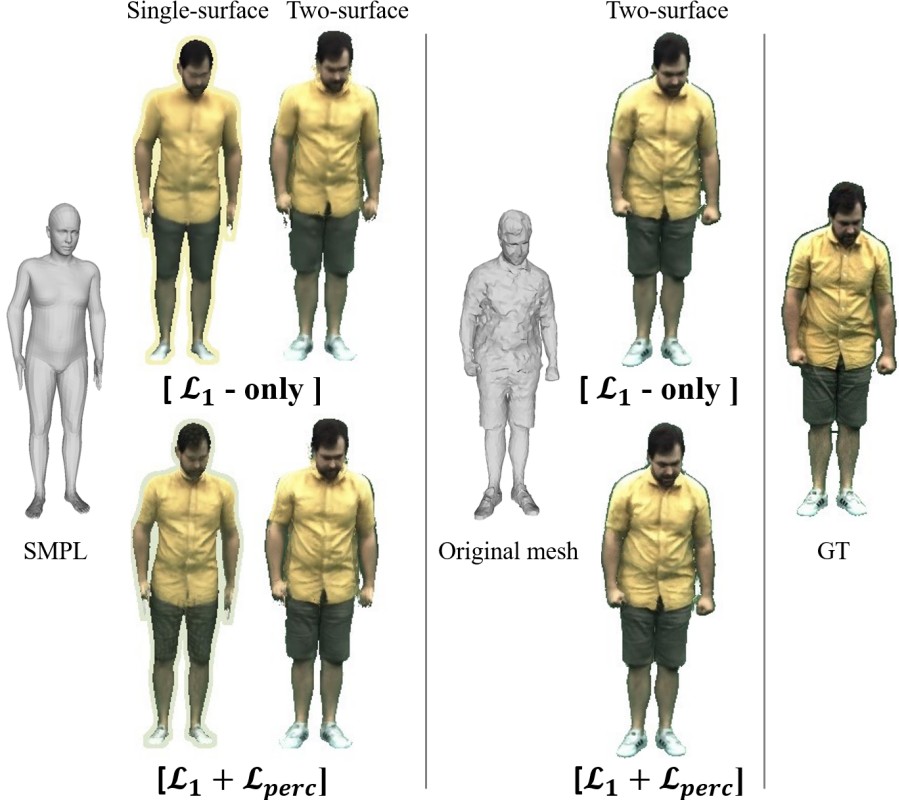

Figure 10: **Ablations** on **using SMPL** as a deformable human model. Evaluated on the novel view synthesis task for the $D_2$ subject. Our method with very coarse SMPL mesh can still recover the real geometry compared to the single surface baseline where the approach can only paint on the given mesh.

## C Reproducibility

### C.1 Implementation Details

**Feature Extractor.** The feature extractor for each surface is based on the same U-Net architecture [19]. We use the feature extractor to encode the temporal normal map $(T + 1) \times 1024 \times 1024 \times 3$ into the $1024 \times 1024 \times 32$ feature map. Here, the temporal normal map is the concatenation of the posed normal maps of the motion window $[f, f - T]$. The feature extractor consists of 10 down-sampling layers (with a down-sampling rate of two) and 10 up-sampling layers (with an up-sampling rate of two). There is a skip connection between every corresponding down and upsampling layer.

**Light Field MLP.** The light field MLP consists of 8 layer with 256 channel size. ReLU activation is used in between the layers. Input to the light field MLP are the positional encodings of two intersection uv coordinates. Then we concatenate the temporal normal map features sampled from the inner and outer surface to the intermediate output of $4^{th}$ layer. Output of the light field MLP is the 3-channel RGB value.

### C.2 Training Details

**Deformable Human Model.** We leverage the deformable human model of Habermann et al. [16], *i.e.*, DDC. The geometry networks of the DDC model are additionally trained with a Chamfer distance supervision with respect to 4D multi-view stereo reconstructions. Originally, we reported the numbers provided by the authors of DDC in Tab. 1. We trained TexNet of DDC with our exact checkpoint for

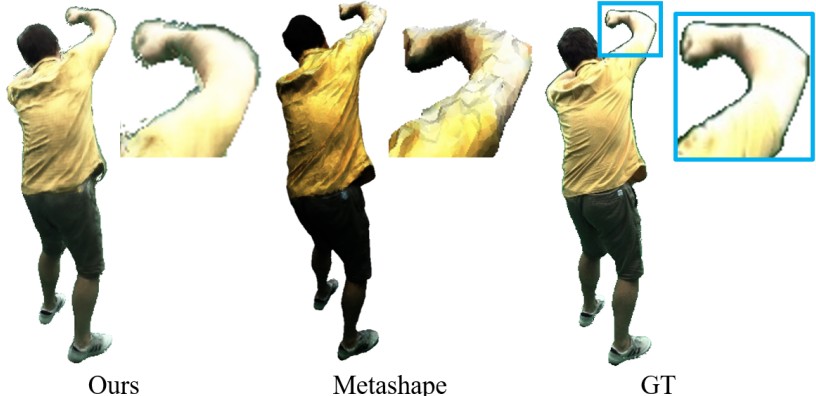

| Ours | Metashape | GT |

Figure 11: **Comparison** with MVS method (Metashape). We compare with the commercial photogrammetry software, Metashape [1]. The reconstructed surface is noisy and the texture exhibits ghosting artifacts. Also, Metashape takes 6 minutes to reconstruct while ours work in real-time ($31fps$). Lastly, our method is animatable, while multi-view stereo methods are not.

| Method | PSNR↑ | LPIPS↓ | FID↓ |
|---|---|---|---|
| DDC (paper) | 32.96 | 20.07 | 27.73 |
| DDC (same geometry network) | 29.50 | 32.66 | 44.18 |
| **Ours** | **33.30** | **12.85** | **8.69** |

| Method | PSNR↑ | LPIPS↓ | FID↓ |
|---|---|---|---|
| DDC (paper) | **28.05** | 30.43 | 38.37 |
| DDC (same geometry network) | 26.59 | 43.43 | 61.92 |
| **Ours** | 27.95 | **26.59** | **26.16** |

(a) *View synthesis error* on *D2*         (b) *Motion synthesis error* on *D2*

Table 5: We trained TexNet of DDC with our exact checkpoint for the geometry networks (EGNet and DeltaNet) that are trained with Chamfer distance to alleviate any other influence. We found that our geometry checkpoint has even a lower performance than the original one. Note that our representation clearly outperforms DDC.

the geometry networks trained with Chamfer distance supervision (EGNet and DeltaNet) to alleviate any other influence. We found that our geometry checkpoint has lower performance than the original one. Note that our representation clearly outperforms DDC (see Tab. 5). Moreover, our ablation 'single surface + viewing direction' (Fig. 5-b,Tab. 2-b) can be considered as a variant of DDC further confirming the superiority of our appearance representation.

**DELIFFAS.** Both the feature extractors and light field MLP are trained in an end-to-end manner using Adam optimizer [25]. We used the learning rate of $2 \times 10^{-4}$. We train on a single RTX 8000 48G GPU with a single batch size. We supervise our approach by minimizing the following loss $\mathcal{L} = \lambda_1 \cdot \mathcal{L}_1 + \lambda_{\mathrm{perc}} \cdot \mathcal{L}_{\mathrm{perc}}$, where $\mathcal{L}_1$ and $\mathcal{L}_{\mathrm{perc}}$ are the $L_1$ loss and perceptual [22] loss, respectively. $\lambda_1$ and $\lambda_{\mathrm{perc}}$ are their weights. For the first $940k$ iterations, $\lambda_1$ is set to one, and $\lambda_{perc}$ is set to zero. Then, we set $\lambda_1$ and $\lambda_{perc}$ so that the magnitude of each loss term is roughly the same and train for additional $350k$ iterations.

### C.3 Runtime at Inference

We report our runtime on the novel pose synthesis task of $D_2$ subject. DELIFFAS runs at $31fps$ for rendering a single $1K$ ($940 \times 1285$) image on a single A100 GPU with an Intel Xeon CPU. The detailed breakdown of runtime is as follows.

Our pipeline can be divided into three stages: (1) obtaining UV (2) feature map generation (CNN) (3) MLP forwarding to get the final color value. **(1) Obtaining UV:** We utilize a GPU accelerated rasterizer to render the image space UV map, which takes 3.58 ms. **(2) Feature map generation (CNN):** Our U-Nets generate normal feature maps in 18.85 ms. **(3) MLP forwarding to compute the final color:** We take 6.78ms to bi-linearly sample the feature corresponding to each pixel, and then the light field MLP takes 2.52ms to compute the color value. In total, DELIFFAS takes 31.73 ms to render a single $1K$ image.

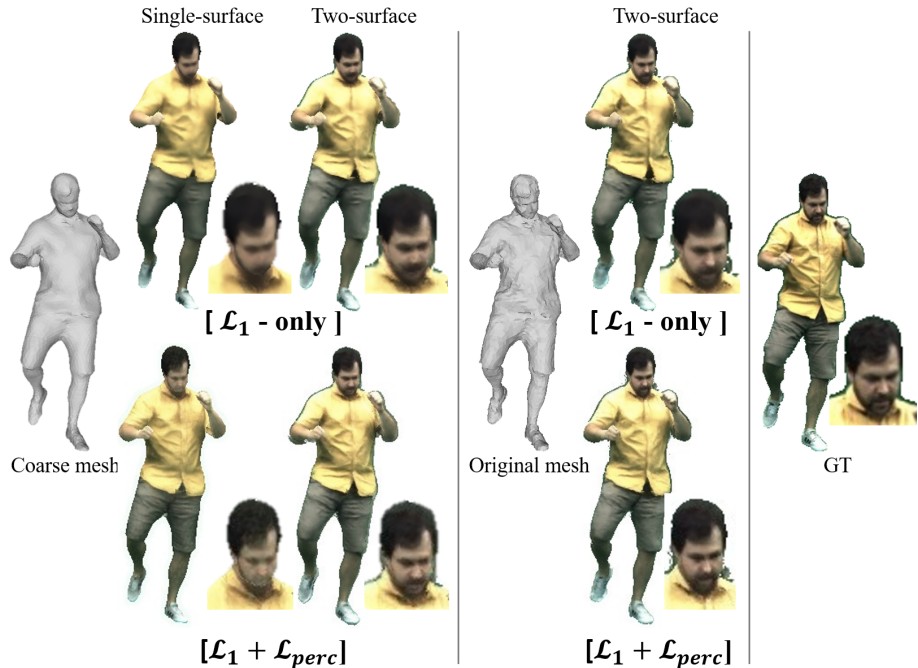

Figure 12: **Ablations** on the deformable template quality **with and without perceptual supervision**. Evaluated on the novel view synthesis task for the $D_2$ subject. Even when using the coarse mesh, our two-surface design enables to generate visual quality similar to the rendering result using original mesh. Employing the perceptual supervision further improves the quality.

Note that we assume the output of the deformable character model, *i.e.*, DDC [16], is already available at training and inference time. DDC is also a real-time method so coupling DDC and our method would not deteriorate the performance much.

We would like to highlight that our method achieves the best of two worlds: runs in real-time as the explicit mesh-based method DDC ($25 fps$), while at the same time achieving superior and comparable visual quality to the hybrid method NA ($0.2 fps$) and HDHumans ($0.3 fps$), respectively.

When tested on an A40 graphics card, we achieve a real-time speed of $26 fps$. Furthermore, most of our computational budget is taken by the U-Net feature map extraction ( 60%) and deterministic bi-linear sampling of features ( 20%). We use a standard U-Net architecture and Tensorflow's bi-linear sampling, but more efficient architecture and sampling implementation can be explored to further reduce the runtime.

## D    Discussion on the Template Mesh Quality

**Rendering Quality.** Our performance degrades if a coarser mesh is used. However, note that our method with coarse geometry or even SMPL mesh can still recover the real geometry compared to the single surface baseline where the approach can only paint on the given mesh (see Fig. 6, Fig. 10, and Fig. 12). Also, it would be an interesting research direction if we could jointly refine the underlying mesh during training and improve the rendering quality.

**Possible Alternatives to the Deformable Human Model.** The deformable human model we used in the main text is DDC [16]. Alternatively, one can use the SMPL model (see Fig. 10) or SMPL+D to deal with the loose clothing. Also, we can use recent avatar NeRF works [64, 21, 42] to compute the canonical mesh (template mesh) and then use skinning-based deformation to obtain the deformed mesh for the new pose.

# E   Societal Impacts

Our research presents potential societal impacts that should be considered. Our method can create avatars using only RGB supervision, which has the potential to democratize immersive VR experiences. Unlike traditional methods that rely on costly setups and specialized professionals, our approach offers accessibility to a wider range of individuals. Additionally, digital doubles generated through our techniques can be used to perform dangerous stunts, reducing risks for human actors. Moreover, individuals can engage in virtual exploration using their personalized avatars.

However, it is important to acknowledge the potential negative consequences associated with our method. The generation of fake media poses a significant risk, potentially leading to public misinformation and eroding trust in media content. We recognize these concerns and emphasize the need for public awareness, responsible use, and disclosure. We strive and strongly hope that our work is applied in ways that positively impact society.

# F   Limitations

Although our method achieves state-of-the-art results in terms of visual quality and runtime, it is not free from limitations. For now, we consider the deformable surface representation is given as an input and our method builds up on it. However, surface deformation and image synthesis can be treated jointly and help to improve each other. Inspired by that, future work could extend our deformable two-surface light field representation such that it can backpropagate into the geometry directly and, thus, potentially refine it throughout the training process. Moreover, the face and hand are not animatable. Extending our work to support the face and hand animation would enhance the versatility of our method and enable many interesting applications such as gaming, and VR. Very complex clothing (*e.g.*, highly glossy garment) might be challenging, though, in contrast to many related works, we demonstrate that loose clothing can work while most other methods only operate under the assumption of tight clothing. The input to our system is a temporal normal map and it is not sufficient enough to fully describe the subject's clothing state which can result in rather blurry images. This is why we employed perceptual supervision so that we can recover the fine details. However, it would be a promising direction to adopt the generative models (*e.g.*, GAN, VAE) to generate even more photorealistic details.

