# OpenReview forum: "DELIFFAS: Deformable Light Fields for Fast Avatar Synthesis"
_NeurIPS.cc/2023/Conference — NeurIPS 2023 poster_

### Official Review · Reviewer_qBLT · 2023-07-05

**Soundness:** 3 good
**Presentation:** 2 fair
**Contribution:** 3 good
**Rating:** 6
**Confidence:** 4

**Summary:**

The paper introduces a method for real-time rendering of animatable full-body avatars. The main contribution is a hybrid mesh-lightfield representation: two surfaces are produced with an off-the-shelf motion-to-surface model, and normals of those surfaces are then used as input to two UNet, the output of those is fit into the lightfield MLP as features. Method is near real-time and runs at 30FPS on a GPU for 1K resolution. Quantitative and qualitative results over recent baselines demonstrate that the method is on par with state-of-the-art.

**Strengths:**

- Paper is well-written and is easy to follow.
- Proposed method seems technically sound, and the two-layer surface formulation fits really well with the lightfield-based neural rendering.
- Apart from the quality, proposed formulation runs at 31fps for a 1K resolution image, and only requires 1 MLP evaluation per ray.
- Quantitative and qualitative evaluation is thorough, the choice of baselines is reasonable (although there are some potential issues, please see below). Ablation study is present and seems to suggest that the proposed two-layer formulation is less sensitive to the quality of the mesh (which can be considered a contribution in itself).


**Weaknesses:**

- There are not too many details on how the deformed mesh is obtained, but assuming it is only conditioned on the motion, it is unclear whether providing normals of the resulting mesh should be sufficient to reconstruct the full image: it would seem that the state of person's clothing cannot be fully described by the pose or sequence of poses.
- On a similar note, it looks like the underlying mesh actually seems to have some measurable effect for the method performance: judging from numerical results in Table 2)? From Figure 6, it is unclear if the quality comparison is done for both models trained with perceptual loss or not? Yet, authors do not really provide any details on how to build the underlying motion-to-mesh model.
- Claim "This allows perceptual supervision ... compared to previous approaches" is incorrect, e.g. [Habermann'15], [Bagautdinov'21], DVA [Remelli'22] should support full-image supervision with arbitrary perceptual losses. Generally, it is a bit unclear how much of the qualitative improvements are coming from the use of perceptual loss (which I would not consider a contribution of this work).

**Questions:**

- Does the mesh model take as input anything apart from the pose? Would it be sufficient to capture non-pose-dependent deformations?
- Do you expect the quality of the method to degrade on grazing angles?
- Is the number of parameters between a single- and two- layered model the same? In the two-layer case, there are two UNets, which could also explain the boost in the performance?
- Could you please confirm that the qualitative comparison (Figure 6) is done consistently for both single- and two-layer models trained with/without perceptual loss? Might be a good idea to be more explicit about this in Table 1 as well.


**Limitations:**

Authors did not really discuss limitations.
Suggestions:
- inability to capture realistic dynamics due to information-defficient inputs.
- artifacts on grazing angles due to incoherent features between the two layers.

---

> ### Author Rebuttal · Authors · 2023-08-10
>
> We thank the reviewer for the valuable feedback to further improve our work. Please see our visual and quantitative ablations in the global response-rebutal.pdf.
>
> * * *
> **Details about the mesh**
>
> The method we use to compute the deformed mesh is DDC [1] and it is only conditioned on the motion history. We will include more details about it in the supplementary material. It is true that the normal map is not sufficient enough to fully describe the subject’s clothing state, which can result in rather blurry images. This is why we employed perceptual supervision so that we can recover the fine details.
>
> * * *
> **Quality of mesh**
>
> Our performance degrades if a coarser mesh is used. However, note that our method with coarse geometry or even SMPL mesh can still recover the real geometry compared to the single surface baseline where the system can only paint on the given mesh (see main paper-Fig.6 and rebuttal.pdf-Fig.2,4). Also, it would be an interesting research direction if we can jointly refine the underlying mesh during training and improve the rendering quality.
> The mesh model we used is DDC. Alternatively, one can use the SMPL model (see rebuttal.pdf-Fig.4) or SMPL+D to deal with the loose clothing. Also, we can use recent Avatar NeRF works to compute the canonical mesh (template mesh) and then use skinning-based deformation to obtain the deformed mesh for the new pose. We will include more details about DDC and other alternatives in the supplementary material.
>
> For Fig.6, both models are trained without a perceptual loss. We included the result only with L1 supervision for the ablations because otherwise it would be unclear whether the proposed two-surface representation or the perceptual supervision improves the result quality. We will make this more clear in the revision. We also included ablations with perceptual supervision in the rebuttal document (see rebuttal.pdf-Fig.1,Fig.2,Tab.1)
>
> * * *
> **Claim about perceptual supervision**
>
> We did not claim that we are the only work that employs the perceptual supervision on the entire image. We claimed that we can employ it thanks to our fast rendering speed while coordinate-based MLP methods (e.g., NeRF methods) cannot or can only employ perceptual supervision on the small patch due to its slow rendering speed. As shown in the supplementary material-Fig.1 and rebuttal.pdf-Fig.1, the perceptual supervision on the entire image allows recovering the fine details.
>
> * * *
> **Input to the mesh model**
>
> The only input to the deformation surface model is the pose history. This is not sufficient to capture fine details, and this is why we employ perceptual supervision to represent such details.
>
> * * *
> **Grazing angles**
>
> We have already shown the rendering results on the grazing angles in the supplementary material video (00:14-00:30, 00:45-00:54, 02:17-02:27), where the camera is freely rotating around the subject. However, we have found no notable artifact or quality degradation on grazing angles.
>
> * * *
> **Number of parameters**
>
> The number of total parameters is larger in the two-surface model as it includes two separate U-Nets for extracting feature map from each surface while the single-surface model only has single U-Net. The U-Net and MLP architecture used in both single and two-surface model are the same.
> To verify that the performance boost is coming from the two-surface design, we additionally trained a “single-surface with two U-Net” variant (see rebuttal.pdf-Fig.5,Tab.1-c,i). Even though now the network size is the same, our two-surface representation still outperforms the single-surface baseline.
>
> * * *
> **Consistent experiments in Figure 6 and Table 1**
>
> The number reported in Table 1 (comparison with other works) is computed with our final model with full supervision (two-surface + L1 + perceptual). The variants in Table 2 (ablation study) are all trained with L1-only supervision except our final model (Tab.2-d). The qualitative results on smooth mesh in Fig.6 is also trained with L1-only supervision for both the single and two-surface models. Again, excluding the perceptual supervision from the ablation study is to verify the effectiveness of our two-surface design. Lastly, we additionally verify the effect of perceptual supervision on the entire image in the supplementary document Fig.1. We will make it more clear in the text and figure caption that the results in Fig.6 are generated with models trained only with L1 supervision.
>
> Also, we included the ablation results with perceptual supervision as well in the rebuttal.pdf-Fig.1,2,Tab.1. Similar to the results without perceptual supervision, our full model outperforms other variants.
>
> * * *
> **Limitations**
>
> Thank you for the suggestion. Regarding the limitation 1, it is true that the normal map is not sufficient enough to fully describe the subject’s clothing state which can result in rather blurry images. This is why we employed perceptual supervision so that we can recover the fine details. However, it would be a promising direction to adopt the generative models (GAN, VAE, …) to generate even more photorealistic details. Regarding the limitation 2, although we have found no notable artifacts on the grazing angles (as can be seen in the free-viewpoint rendering result in the supplementary video), it would be an interesting research direction to further refine the designs for the corner cases. We will include these in the limitations section of the revision.
>
> [1] Habermann et al. “Real-time deep dynamic characters.” In TOG 2021.

---

> > ### Comment · Reviewer_qBLT · 2023-08-18
> >
> > Thanks for the detailed response. I would encourage authors to reformulate the writing to make the point about perceptual loss more clear. I think fundamentally only conditioning on pose history is not enough no recover the true underlying details, but I do see that perceptual metric is a better choice than just L2 to tackle this (although adversarial training would make more sense tbh) - but not sure if this is a particularly novel observation which should be claimed as a contribution.
> > Overall though, I think the two-layer lightfield representation proposed in this work is a good idea, and thus I stand by my original rating.

---

> > > ### Author Response · Authors · 2023-08-19
> > > **Response to Reviewer qBLT**
> > >
> > > Thank you for taking your time to review our work. We will make the claim about perceptual supervision more clear in the final version. Moreover, we agree that modeling the non-pose dependent effects more explicitly would be an interesting venue for future work.

---

### Official Review · Reviewer_LHHA · 2023-07-06

**Soundness:** 3 good
**Presentation:** 3 good
**Contribution:** 3 good
**Rating:** 5
**Confidence:** 4

**Summary:**

This work proposes a method for human avatar reconstruction from multiview video data. This work leverages deformable light fields to model the geometry and texture. Experiments show that this method outperforms the existing methods in terms of novel view and novel pose synthesis.

**Strengths:**

1. The idea of using light field to improve computation efficiency is novel and well-motivated. It can overcome some of the drawbacks of NeRF, such as the heavy computation cost for point sampling.
2. Using mesh as geometry representation is also reasonable as it is naturally compatible with conventional rendering engines, which could open up many application possibilities.

**Weaknesses:**

1. Although the framework is novel, the idea of using differentiable rasterization for animatable human avatar creation has already been explored in prior work[1], please explain the difference between this work and [1].
2. There may exist a risk of overfitting because this work only experiments on one dataset, it could be better to add more experiments on the commonly used benchmark dataset such as ZJU-Mocap.
3. Efficiency is one of the most important contributions of this work, the authors could report more numerical results to verify its rendering speed, and it’s better to compare it with [1].
4. The setting for ablation study (line 269) is confusing, I don’t understand why other variants are not trained with perceptual loss. I think this loss term should not be skipped except for studying the necessity of optimization objectives.

[1] UV Volumes for Real-time Rendering of Editable Free-view Human Performance

**Questions:**

The training data are multiview videos, is the proposed method possible to be trained on monocular videos?

**Limitations:**

Addressed.

---

> ### Author Rebuttal · Authors · 2023-08-10
>
> We thank the reviewer for the valuable feedback to further improve our work. Please see our visual and quantitative ablations in the global response-rebutal.pdf.
>
> * * *
> **Difference to UV volumes**
>
> Please note that Chen et al. [1] is a concurrent work presented in CVPR 2023, which was held after the NeurIPS submission. The difference between [1] and ours is as follows: [1] is a single surface-based method that is tightly bounded to the underlying mesh (optimized uv map). However, ours can go beyond the underlying mesh and recover the real geometry that lies between the two surfaces (see main paper-Fig.5,6, rebuttal.pdf-Fig.2,4). Also, [1] supervises the uv map with the DensePose prediction. Therefore, as mentioned in their limitation section, they can only handle clothing types that roughly fit the human body. On the other hand, ours can handle loose garments such as skirts. We will cite this work and discuss the differences in the revised version.
>
> * * *
> **Risk of overfitting**
>
> DynaCap is also a well-established benchmark used by other works [2,3,4,5] and known to be significantly more challenging than ZJU-Mocap in terms of pose variety. For example, the test sequence comprises around 7000 frames with challenging poses. Further, our results show that methods that perform reasonably well on ZJU-Mocap obtain significantly more blurred results on DynaCap due to the increased pose variety (see main paper-Fig.4 NB, A-NeRF results).
>
> * * *
> **Numerical results for runtime performance**
>
> We report the performance comparison with Chen et al. on a single A100 GPU.
> Due to the limited time, we report the performance of Chen et al. according to their paper-Tab.4.
> Chen et al. achieves 18 fps on the novel pose synthesis task while ours achieves 31 fps. Although there are some differences in the exact pipeline, the pipeline of Chen et al. and ours can be divided into three stages: (1) obtaining UV (2) feature map generation (CNN) (3) MLP forwarding to get the final color value.
>
> **(1) Obtaining UV**
>
> Chen et al. generates UV feature volume using sparse CNN (48.78ms). Density along the camera ray is computed (7.08ms). Then the image space uv feature map is volume-rendered (1.73ms). The UV mlp converts the feature into uv coordinates (1.53ms). This stage sums up to 59.12 ms.
>
> We utilize GPU accelerated rasterizer to render the image space uv map, which takes 3.58 ms.
>
> **(2) Feature map generation (CNN)**
>
> Chen et al. takes 7.52ms to compute the neural texture map.
>
> Our U-Nets generate normal feature maps in 18.85 ms.
>
> **(3) MLP forwarding to compute the final color**
>
> Chen et al. takes 1.60 ms to compute color using MLP.
>
> We take 6.78ms to bi-linearly sample the feature corresponding to each pixel. And then the light field MLP takes 2.52ms to compute the color value.
>
>
> In total, Chen et al. takes 68.23 ms and ours take 31.73 ms. We will include this comparison in the revision.
>
> * * *
> **Ablation study**
>
> The reason why we included the ablation results generated only with L1 supervision is because otherwise it would be unclear whether the proposed two-surface representation or the perceptual supervision improves the result quality.
> We also included the ablation results with perceptual supervision as well in the rebuttal.pdf-Fig.1,2,Tab.1. Similar to the results without perceptual supervision, our full model outperforms other variants.
>
> * * *
> **Monocular videos**
>
> Although we could not try due to the limited time, we expect that the performance would naturally degrade as the system has less chance to observe the multi-view information. We will leave this as future work.
>
> [1] Chen et al. “UV Volumes for Real-time Rendering of Editable Free-view Human Performance.” In CVPR 2023.
>
> [2] Habermann et al. “Real-time deep dynamic characters.” In TOG 2021.
>
> [3] Liu et al. “Neural actor: Neural free-view synthesis of human actors with pose control.” In TOG 2021.
>
> [4] Zheng et al. “Structured Local Radiance Fields for Human Avatar Modeling.” In CVPR 2022.
>
> [5] Peng et al. “Implicit Neural Representations with Structured Latent Codes for Human Body Modeling.” In TPAMI 2023.

---

> > ### Comment · Reviewer_LHHA · 2023-08-18
> >
> > Thank the authors for the clarifications. The additional numerical results have addressed most of my concerns about computation cost and ablation studies. However, the risk of overfitting has not been solved yet. I appreciate that the authors named some prior works [2,3,4,5] to prove that DynaCap is a widely used benchmark, and I agree with this. But I carefully checked [2,3,4,5] and find that all of them experiment on two or three different datasets or at least sample sequences from different datasets to avoid overfitting. Unfortunately, there are no results for the second dataset (neither quantitative nor qualitative) in the rebuttal. Thus, I will keep my rating.

---

> > > ### Author Response · Authors · 2023-08-19
> > > **Response to Reviewer LHHA**
> > >
> > > We thank the reviewer for the feedback and are happy to hear that most concerns are addressed by our rebuttal, which we will also carefully incorporate in our final version.
> > >
> > > Concerning the overfitting, we agree that an additional sequence would further confirm the superiority of our design. Though, due to the limited time, it was not possible to obtain such an evaluation. However, we are happy to add more results of other datasets in the final version if requested.
> > >
> > > For now, we would like to again highlight the complexity of the DynaCap dataset compared to other datasets, which makes overfitting nearly impossible. For example, DynaCap training sequences range about 17,000-19,000 frames, and testing sequences range about 7,000 frames including a variety of dynamic and challenging poses. And this thus already validates our method. Please note that ZJU-Mocap which the reviewer suggested only ranges about 300-1,000 frames, which is much less challenging. Moreover, we already show qualitative results on other subjects, which include a subject with challenging loose garments (i.e., skirts). This also qualitatively confirms our generalizability.
> > >
> > > As the reviewer pointed out, we have addressed the reviewer’s other concerns. For the only concern about the evaluated datasets, we believe that the DynaCap dataset is more challenging in terms of motion diversity compared to other datasets and our improvements on the DynaCap dataset prove the superiority of our method. Considering these, we hope you could consider increasing your rating.

---

> > > > ### Comment · Reviewer_LHHA · 2023-08-20
> > > >
> > > > Thank the authors for the response. I want to further explain my concern about overfitting. Overfitting does not only means the method cannot generalize to the new subjects in DynaCap or cannot handle loose clothing. It also means it's possible that this method works well on DynaCap (as the authors mentioned, it has more training frames, more viewpoints, and diverse pose distribution) but struggles to gain convergence on other datasets which contain less diverse training data. And this is why I also asked the question about monocular video in my initial review.
> > > >
> > > > Considering the standard of NeurIPS and prior works, I believe this point is very important and can help readers understand the superiority as well as the limitations of the proposed method. It is highly recommended to include the results of other datasets in the final version. I would like to increase the rating if the authors agree to add these experiments.

---

> > > > > ### Author Response · Authors · 2023-08-20
> > > > > **Response to Reviewer LHHA**
> > > > >
> > > > > We thank the reviewer for the insightful and valuable feedback.
> > > > >
> > > > > We promise to add the comparisons on ZJU-Mocap dataset in the final revision.
> > > > >
> > > > > We are already working on it but it is not possible to update the new results by the discussion period deadline (Aug 21st) since we are currently in the process of converting ZJU-Mocap dataset into our system format (i.e., converting the ZJU-Mocap camera parameters and SMPL vertices into the same metric that we use, converting the SMPL mesh vertices into our mesh reader file format, fixing the evaluation protocol to follow Neural Body and etc). We hope to have the results soon and we will make sure to include them in the final revision.
> > > > >
> > > > > We will include comparison results on ZJU-Mocap following the evaluation protocol of Neural Body specified in their Github page [1], which is as follows:
> > > > >
> > > > > 1. Subjects: 313, 315, 377, 386, 387, 390, 392, 393, 394
> > > > > 2. Train / test camera split: Out of 23 cameras, [0, 6, 12, 18]-th camera will be used for training and the remaining will be used for testing.
> > > > > 3. Frame split: we will use the training frames for the novel view synthesis task and held-out frames for the novel pose synthesis task.
> > > > >
> > > > > Finally, we will also provide an ablation on the number of cameras used for training in the final revision, including the monocular case as well.
> > > > >
> > > > > [1] https://github.com/zju3dv/neuralbody/blob/master/supplementary_material.md

---

### Official Review · Reviewer_cQc3 · 2023-07-07

**Soundness:** 2 fair
**Presentation:** 2 fair
**Contribution:** 2 fair
**Rating:** 6
**Confidence:** 3

**Summary:**

The paper introduces DELIFFAS, an innovative method for generating controllable and photorealistic digital human avatars in real-time. This system utilizes a deformable two-surface representation to parameterize a surface light field, deforming two surfaces according to a deformable mesh model, allowing the light field to be driven by skeletal motion. This approach significantly increases inference speed, enabling the entire image to be rendered during training, unlike previous methods. This faster rendering time also allows for full-image level perceptual supervision, in contrast to earlier approaches, which could only supervise individual pixels or small patches due to slow runtime.

**Strengths:**

Key strengths of the paper:

- The proposed method DELIFFAS can generate controllable and photorealistic human avatars in real-time.

- The use of a deformable two-surface representation to parameterize a surface light field improves both accuracy and efficiency of appearance synthesis.

- The efficient design of the neural architecture and light field representation allows for integration into the graphics pipeline, resulting in real-time performance.

- Due to the high inference speed, the system can render the entire image during training and employ full-image level perceptual supervision.

**Weaknesses:**

Some potential weaknesses of the paper:

- Limitations in Pose Variety: It is unclear how well the model would perform with extremely unusual or rare poses, or if it would be able to accurately generate and maintain details under these conditions.

- Dependency on Underlying Mesh Quality: While the authors have demonstrated that their method can tolerate coarser geometry, the accuracy and realism of the output may still depend on the quality of the underlying mesh.

- Robustness to Complex Clothing:  While the authors mention that their method can handle "challenging loose garment types," there may be limitations when dealing with more complex or unusual clothing, especially if such examples are not well represented in the training data. This can be often the case where common clothing could exhibit highly glossy or specular lighting effects such as silk.

**Questions:**

- How does the DELIFFAS model perform when exposed to extreme poses or movements not well-represented in the training data? Are there specific types of movement or pose it struggles to accurately represent?

- Could you elaborate on the computational resources required for real-time implementation of the DELIFFAS model? A100 is a quite heavy GPU configuration, what happens if tested on other GPU configurations for the performance?

-  How would the system handle more complex or unusual clothing types, especially those not well-represented in the training data?

- The method seems to rely heavily on the quality of the underlying mesh model.

**Limitations:**

Discussed above, in the weakness section.

---

> ### Author Rebuttal · Authors · 2023-08-10
>
> We thank the reviewer for the valuable feedback to further improve our work. Please see our visual and quantitative ablations in the global response-rebutal.pdf.
> * * *
> **Pose variety**
>
> Typically our method is robust to rather challenging poses, however, completely out-of-distribution poses such as handstand might not work since cloth deformation can be very large, i.e., shirt pulls down. We encourage the reviewer to look at the publicly available test set of the DynaCap sequences, which comprise a large variety of poses.
>
> * * *
> **Dependency on mesh quality**
>
> Our performance can degrade if a coarser mesh is used. However, note that our method with coarse geometry or even SMPL mesh can still recover the real geometry compared to the single surface baseline where the approach can only paint on the given mesh (see main paper-Fig.6, rebuttal.pdf-Fig.2,4).
> Also, it would be an interesting research direction if we can jointly refine the underlying mesh during training and improve the rendering quality. However, it is important to note that the deformable character model is also solely learned from multi-view video, thus, input assumptions are the same compared to competing methods.
>
> * * *
> **Robustness to complex clothing**
>
> Very complex clothing (e.g., highly glossy garment) might be challenging, though, in contrast to many related works, we demonstrate that loose clothing can work while most other methods only operate under the assumption of tight clothing. We will discuss this in the limitations.
>
> * * *
> **Computational resources**
>
> As we stated in the main paper L41-45, our method runs at 31 fps for rendering a single 1K image (940X1285) on a single A100 GPU with Intel Xeon CPU.
> When tested on A40, we achieve the real-time speed of 26 fps. Also, please note that our method runs X100 faster than the SOTA methods NA and HDHuman when the same computing resource is given.
>
> Furthermore, most of our computational time is coming from the U-Net feature map extraction (\~60%) and deterministic bi-linear sampling of features (\~20%). We use a standard U-Net architecture and Tensorflow’s bi-linear sampling, but more efficient architecture and sampling implementation can be explored to further reduce the runtime.
>
> * * *
> **Clothing not in training data**
>
> Our method is a person/clothing specific method and we do not claim generalization to new clothing (as most previous work does).

---

### Official Review · Reviewer_dLaT · 2023-07-07

**Soundness:** 3 good
**Presentation:** 3 good
**Contribution:** 2 fair
**Rating:** 5
**Confidence:** 5

**Summary:**

This paper aims to learn digital human body avatars from multi-view videos. To achieve both photorealism and fast inference speed, the authors introduce a new representation based on a surface light field. The surface light field is attached to a drivable human mesh model, and is conditioned on skeleton motion. Such a representation can be easily defined on texture atlas, and efficiently rendered with standard rasterization pipelines. Overall, the proposed representation allows perceptual supervision and real-time rendering, resulting in superior performance over existing baselines.

**Strengths:**

1. The proposed method enables real-time rendering of full-body human avatars with a resolution of 1K. This is a nice property for many down-stream interactive applications.

2. The proposed representation further allows perceptual supervision on the full image, which is crucial for learning high-quality appearance details like cloth wrinkles.

3. All important implementation details are adequately discussed in the main paper and the supplemental document. For example, in line 179-180, the authors describe how they handle corner cases where a ray only intersects with the outer surface.

**Weaknesses:**

1. Although the proposed representation is based on a light-field defined by two deformable surfaces and demonstrates superior performance over the single-surface baseline Sec.4.3, it is still tightly bounded to the underlying mesh model. According to Line 149, the shell bounded by two surfaces has a thickness of only 3 cm, which is relatively small. This means that the underlying mesh model should be a close approximation of the real avatar surface. Therefore, I am suspicuous whether other human body templates (like SMPL) can be used in the proposed method, although the authors claim in Line 122 that "any other deformable surface representation could be used as well".


2. Although the proposed two-surface representation is nice for modeling dynamic texture, the geometric deformation is not modeled. In Line 118, the authors mention that they assume "a skeletal motion-depdent deformable mesh surface is given". This can be obtained for the task of novel view synthesis, as mesh tracking is a key step in data preprocessing. However, this is very challenging for novel pose synthesis, especially for loose garments like a long dress. Unfortunately, I can't find any details about how they obtain the mesh model for novel poses in the paper.


3. The authors claimed in Abstract that the combination of photorealism and inference speed "still remains unsolved", which I think is not true. Some recent works have already achieved real-time neural character rendering with high photo-realisitc quality. For example, Ouyang et al. [a] proposed to use multi-plane image (MPI), which is also a light field representation, to render an animatable character in real-time. Ouyang et al. also applied perceptual supervision on full images to learn high-quality appearance details. Therefore, I think this submission should cite, discuss, and perhaps compare with this highly relevant work [a].

[a] Ouyang et al. Real-Time Neural Character Rendering with Pose-Guided Multiplane Images. ECCV 2022.


4. In Line 199-200, the authors mention that the blurry results trained with L1 loss is due to the one-to-many mapping problem. However, their solution, i.e., applying perceptual supervision, does not resolve this problem; in other words, the one-to-many problem still exists after applying the perceptual loss. I guess this is the main reason for the jittering texture in the supplemental video (00:14-00:29).

5. The novel pose sequences for the long skirt case (supplemental video 00:56-00:59) are too short.


Typos: Line 278: In Fig.5, **We** --> In Fig.5, **we**


**Questions:**

None

**Limitations:**

The authors discuss the limitations and potential societal impact in the supplemental document.

---

> ### Author Rebuttal · Authors · 2023-08-10
>
> We thank the reviewer for the valuable feedback to further improve our work. Please see our visual and quantitative ablations in the global response-rebutal.pdf.
> * * *
> **Bounded to mesh**
>
> We conducted an experiment where we use the SMPL model as the deformable mesh model (see rebuttal.pdf-Fig.4). One can see that our approach even for this very coarse model still outperforms the baselines proving the flexibility of our representation while a better template mesh further improves the results. We will include this experiment in the camera ready version.
> * * *
>
> **Details about the mesh model**
>
> We used the off-the-shelf DDC [1] as our deformable surface model as mentioned in Section 3.1. This model takes a (novel) skeletal motion as input and generates the posed and non-rigidly deformed template while being supervised solely on multi-view imagery. We will add more details about DDC in the final version.
>
> * * *
> **Claim with respect to Ouyang et al.**
>
> Although Quyang et al. achieve impressive and photorealistic results, they only allow minimal pose changes and mostly overfit to the specific recording as also stated in their limitations. In contrast, our method can handle arbitrary poses at test time and, thus, is a controllable character representation. Moreover, they mostly show frontward facing camera movements, while our method supports full 360 camera rotations around the human subject. We will adjust the claims while considering Quyang et al. and discuss the differences between Quyang et al. and ours in the revision.
>
> * * *
> **Solving the one-to-many mapping**
>
> Like most works we do not explicitly model these stochastic effects in clothing, but we found that the perceptual loss effectively addresses them (see main paper-Fig.5 c,d) while some small residual might still remain. Also, since we can render the entire image thanks to the fast rendering speed,  generative models (e.g., GAN, VAE) could be incorporated to model this stochastic process. We leave this as future work.
>
> * * *
> **Sequence too short**
>
> We will increase the length of the sequence.
>
> [1] Habermann et al. “Real-time deep dynamic characters.” In TOG 2021.

---

> > ### Comment · Reviewer_dLaT · 2023-08-20
> >
> > I would like to thank the authors for their reply to my questions. My major concerns are addressed in the rebuttal. After reading the rebuttal and other reviews, I would like to keep my original positive position.

---

### Official Review · Reviewer_QJHp · 2023-07-07

**Soundness:** 3 good
**Presentation:** 3 good
**Contribution:** 3 good
**Rating:** 7
**Confidence:** 5

**Summary:**

The paper presents a real-time approach for generating controllable and photorealistic digital human avatars. The main contribution of the paper is to learn the texture of the avatar using a light-field representation. In contrast to other continuous representations (e.g., NeRF), the proposed method can predict the color of a ray using a single MLP call. This allows faster rendering of the images which in turn allows using perceptual losses during training and yields significantly better synthesis results. The paper mainly extends the Deep Dynamic Characters (DDC) method and replaces its TextureNet with the proposed light-field representation. The experiments are performed on the DynaCap dataset where the proposed method is shown to outperform other real-time methods while being on par with the non-realtime methods.

**Strengths:**

- The use of the light slab method for learning the texture details of the avatar is novel and makes sense. I believe this has the potential to be used in other methods as well where faster rendering speed is required.
- Since the color for each ray can be predicted in one shot, the proposed method allows rendering full images during training. Hence, perceptual losses can be used and are shown to help significantly (LPIPS: 18.51 vs 12.85)
- The proposed method achieves results on par with the state-of-the-art while being 100x better.
- The paper is well-written and easy to read, though some typos need to be fixed

**Weaknesses:**

### Novelty
- My main concern with the paper is its limited novelty. The paper builds on DDC and mostly replaces its TextureNet with the proposed light-field-based MLPs. It is unclear whether better results are due to better engineering or the proposed light-field-based approach. Is the implementation of DDC and the proposed method exactly the same with texture nets being the only difference between the two?

### Missing Experiments
- I would have liked to see an experiment in which light-field is replaced with vanilla NeRF while having everything else exactly the same.



**Questions:**

### Checkerboard artifacts:
- What are those checkerboard artifacts in Fig. 4? With such artifacts it is very easy to identify that the images are computer generated, hence it raises a question regarding paper's claim of `photorealistic` synthesis. Perhaps tone down the photorealistc claim if these artifacts are not preventable.

### Experiments
- The best setting of the proposed method is with $L_{perc}$, I wonder why the ablation studies are performed without this loss? What happens if we use $L_{perc}$ with `Single surface + viewing direction` setting?

### Overview figure
Why the red dot is at a different location in the feature maps? Shouldn't it be consistent with the temporal normal maps?


**Limitations:**

The limitations of the paper are not highlighted in the paper. The authors should add the during the rebuttal.

Potential limitations are:
1. Requirement of an initial 3D scan of the person as required by DDC, which is not easy to obtain without having access to sophisticated hardware.
2. Long training sequences.
3. Use in deep fakes.
4. Privacy concerns, etc.

---

> ### Author Rebuttal · Authors · 2023-08-10
>
> We thank the reviewer for the valuable feedback to further improve our work. Please see our visual and quantitative ablations in the global response-rebutal.pdf.
> * * *
> **Novelty**
>
> We would like to emphasize that this work is about finding an appearance representation that is, both, efficient and supports high quality.  To this end, we present the deformable two-surface light field parameterization. We assume a deformable mesh model is given, this can be either DDC or as our ablation shows also a coarser deformable mesh or even SMPL mesh would work reasonably well (see rebuttal.pdf-Fig.2,4). Originally, we reported the numbers provided by the authors of DDC. For the rebuttal, we trained DDC’s TexNet with our exact checkpoint for the geometry networks (EGNet and DeltaNet) to alleviate any other influence. We found that our geometry checkpoint has even a lower performance than the original one. Note that our representation clearly outperforms DDC’s results (see below). Moreover, our ablation ‘single surface + viewing direction’ (main paper Fig.5-b,Tab.2-b) can be considered as a variant of DDC further confirming the superiority of our appearance representation.
>
> | Novel view                  | PSNR      | LPIPS     | FID      |
> |-----------------------------|-----------|-----------|----------|
> | DDC (paper)                 | 32.96     | 20.07     | 27.73    |
> | DDC (same geometry network) | 29.50     | 32.66     | 44.18    |
> | **Ours**                    | **33.30** | **12.85** | **8.69** |
>
> | Novel pose                  | PSNR      | LPIPS     | FID      |
> |-----------------------------|-----------|-----------|----------|
> | DDC (paper)                 | **28.05**     | 30.43     | 38.37    |
> | DDC (same geometry network) | 26.59     | 43.43     | 61.92    |
> | **Ours**                    | 27.95 | **26.59** | **26.16** |
>
>
>
> * * *
> **NeRF ablation**
>
> This might be challenging as rendering the 1K image with volumetric rendering takes very long, e.g., Neural Actor and HDHuman take 4~5 seconds, which makes the perceptual supervision on the entire image inefficient.
>
> * * *
> **Checkerboard artifacts**
>
> This is not a checkerboard artifact. The D2 subject has a stripe pattern on the T-shirt if closed up (see GT of supplementary document-Fig.1 and rebuttal.pdf-Fig.3) and our approach learned to reproduce it. This actually shows our ability to capture very fine details.
>
> * * *
> **Why ablation study without L_perc? / Additional Ablation “single surface + viewing direction + perceptual loss”**
>
> We conducted this ablation to demonstrate that the proposed two-surface representation in isolation with the perceptual supervision achieves improved visual accuracy, which is confirmed by our experiment. Adding the perceptual supervision further improves visual quality.
> For the rebuttal, we further added ablations with perceptual supervision as requested by the reviewer (see rebuttal.pdf-Fig.1,2,Tab.1). The variant with “single surface + viewing direction + perceptual loss” is shown in rebuttal.pdf-Fig.1-e and Tab.1-h.
> In summary, the combination of our proposed two-surface representation and the perceptual supervision achieves the best result.
>
> * * *
> **Limitations**
>
> We will include the suggested limitations in the revision. Regarding the first limitation (requirement of an initial 3D scan which is not easy to obtain), we would like to note that this assumption is not too restrictive as competing methods (we compare to) rely on a multi-view sequence. A template from multi-view imagery could be acquired by methods like multi-view PIFu.
> Regarding the second limitation (long training sequences), this is rather a pro than a con. Most methods are not even able to train on such long sequences since their representation fails to optimize parameters that can cover such diverse poses and as a consequence produce very blurry results (see main paper-Fig.4 NB, A-NeRF results).
> Concerning deep fake generation and privacy concerns, we will add such a discussion.
> Moreover, we would like to highlight that we discuss limitations in the supplemental document. However, for the final version we will incorporate this paragraph into the main manuscript.
>
> ***
> **Overview figure**
>
> The location of the red dots should be consistent with the temporal normal maps. We will modify this in the revision.

---

> > ### Comment · Reviewer_QJHp · 2023-08-21
> >
> > Thank you for providing additional explanations. All my concerns have been addressed. I maintain my original rating.

---

### Author Rebuttal · Authors · 2023-08-10

We thank all the reviewers for their time and effort to review our work and to further improve it. We address individual concerns in the reviewer-specific rebuttal dialogue and provide a pdf (rebuttal.pdf) comprising additional visualizations and experiments.

---

### Comment · Area_Chair_3Wae · 2023-08-20
**The surface light field representation**

Dear Authors of Manuscript 3760 and Reviewers,

In the course of my review, particularly while examining the "360° Viewpoints" section (Sec. 3.2), I noted a potential correlation between the proposed parameterization and the 'surface light field' concept discussed in [1]. The current manuscript does not cite [1], which makes it challenging for me to fully evaluate this aspect. I would appreciate it if the authors could provide further clarity on the distinctions and connections between their work and [1].

[1] Wood, Daniel N., Daniel I. Azuma, Ken Aldinger, Brian Curless, Tom Duchamp, David H. Salesin, and Werner Stuetzle. "Surface light fields for 3D photography." In Seminal Graphics Papers: Pushing the Boundaries, Volume 2, pp. 487-496. 2023.

Kind regards,

AC

---

> ### Author Response · Authors · 2023-08-20
> **Response to AC**
>
> We thank AC for the thoughtful and constructive suggestion to further improve our work.
>
> [1] proposes a surface light fields, which maps a point on the base mesh and viewing direction to the radiance of the real geometry. [1] can encode detailed surface texture and view-dependent effects such as rapid changes in specularity and interreflections, and achieve photorealistic novel view synthesis results.
>
> Both [1] and ours can represent full 360-degree ray space with the light fields defined around the base mesh (surface). However, there are two major differences between [1] and ours.
>
> First of all, [1] is bounded to the underlying base mesh while ours can go beyond the base geometry to recover the real geometry lying between the inner and out surfaces. Specifically, [1] can be thought of as the single-surface light fields conditioned on the viewing direction. Although it can represent more geometric details than are actually present (like bump-mapped lighting), it cannot go beyond the surface of the base mesh to recover the real geometry. On the other hand, our two-surface parameterization allows to not only represent the view-dependent effect but also recover the real geometry lying between the two surfaces. This is why the 'single-surface + viewing direction' variant (main paper Tab.2-b, Fig.5-b) performs better than the 'single-surface' variant (Tab.2-a, Fig.5-a), but performs worse than our two-surface model (Tab.2-c, Fig.5-c). The effectiveness of our two-surface representation in recovering the real geometry is also presented in the results with coarse mesh and even SMPL model (main paper Fig.6, Tab.2 and rebuttal.pdf Fig.2, Fig.4). Please note that the ablations in the main paper are done only with L1-supervision to disentangle the effect of perceptual supervision and prove the effectiveness of our two-surface design. For the ablation results with full L1+perceptual supervision, please refer to rebuttal.pdf-Tab.1 and Fig.1, which also verify the superiority of our two-surface approach.
>
> Moreover, [1] can only represent the static scene while ours can represent and control the dynamic scene, i.e., the human. This is because our light fields are defined on the deformable surface. Also, additional conditioning on the motion history (i.e., temporal normal maps) allows us to represent motion-dependent deformations such as changes in wrinkles.
>
> In summary, both [1] and ours can parameterize full 360-degree ray space by defining the light fields around the mesh surface. However, while [1] is a single surface-based method that is bounded to the underlying mesh surface and can only represent the static scene, our two-surface approach can recover the real geometry lying between the inner and out surfaces and can represent and control the dynamic scene (i.e., human).
>
> We will cite [1] and include the above discussion in the final revision.
>
> [1] Wood, Daniel N., Daniel I. Azuma, Ken Aldinger, Brian Curless, Tom Duchamp, David H. Salesin, and Werner Stuetzle. "Surface light fields for 3D photography." In Seminal Graphics Papers: Pushing the Boundaries, Volume 2, pp. 487-496. 2023.

---

> > ### Comment · Area_Chair_3Wae · 2023-08-21
> >
> > Thanks for the clarification!

---

### Decision · Program_Chairs · 2023-09-21

**Decision:**

Accept (poster)

**Comment:**

The manuscript received uniformly positive evaluations from all reviewers. Although initial concerns were raised regarding incremental adjustments to the DDC framework, the influence of underlying geometry, and the experimental demonstrations, these issues were satisfactorily addressed during the rebuttal phase. Consequently, the AC recommends the acceptance of the manuscript.